# PI3Kα inhibition blocks osteochondroprogenitor specification and the hyper-inflammatory response to prevent heterotopic ossification

José Antonio Valer[1†], Alexandre Deber[1†], Marius Wits[2], Carolina Pimenta-Lope[1], Marie-José Goumans[2], Jose Luis Rosa[1], Gonzalo Sánchez-Duffhues[2,3], Francesc Ventura[1*]

[1]Departament de Ciències Fisiològiques, Universitat de Barcelona, IDIBELL, C/ Feixa Llarga s/n 08907 Hospitalet de Llobregat, Barcelona, Spain; [2]Department of Cell and Chemical Biology, Leiden University Medical Center, Leiden, Netherlands; [3]Nanomaterials and Nanotechnology Research Center (CINN-CSIC), Health Research Institute of Asturias (ISPA), Oviedo, Spain

*For correspondence: fventura@ub.edu

†These authors contributed equally to this work

## eLife Assessment

This study, which includes additional experiments in response to the reviewer comments, presents **valuable** findings illustrating the role of PI3Kα in heterotopic ossification in FOP model mice. The methods, data, and analyses are **solid** and generally support the claims although as noted by one of the reviewers, there is no data demonstrating the effect of BYL79 on cell growth, and it remains unclear whether BYL79 also inhibits the Smad2/3 pathway. While this study provides new insights into the role of the PI3Kα pathway as a therapeutic target for FOP, questions about the mechanism of BYL79 still exist.

**Abstract** Heterotopic ossification (HO) occurs following mechanical trauma and burns, or congenitally in patients suffering from fibrodysplasia ossificans progressiva (FOP). Recently, we demonstrated that inhibitors of phosphatidylinositol 3-kinase alpha (PI3Kα) may be a useful therapy for patients undergoing HO. In this study, using the already marketed BYL719/Alpelisib/Piqray drug, we have further confirmed these results, detailed the underlying mechanisms of action, and optimized the timing of the administration of BYL719. We found that BYL719 effectively prevents HO even when administered up to 3–7 days after injury. We demonstrate in cell cultures and in a mouse model of HO that the major actions of BYL719 are on-target effects through the inhibition of PI3Kα, without directly affecting ACVR1 or FOP-inducing ACVR1[R206H] kinase activities. In vivo, we found that a lack of PI3Kα in progenitors at injury sites is sufficient to prevent HO. Moreover, time course assays in HO lesions demonstrate that BYL719 not only blocks osteochondroprogenitor specification but also reduces the inflammatory response. BYL719 inhibits the migration, proliferation, and expression of pro-inflammatory cytokines in monocytes and mast cells, suggesting that BYL719 hampers the hyper-inflammatory status of HO lesions. Altogether, these results highlight the potential of PI3Kα inhibition as a safe and effective therapeutic strategy for HO.

## Introduction

Heterotopic ossification (HO) is a disorder characterized by ectopic bone formation at extraskeletal sites, including skeletal muscle and connective tissues. Trauma-induced HO develops as a common post-operative complication after orthopedic surgeries (e.g. hip arthroplasty). Blast gene-induced HO, fibrodysplasia ossificans progressiva (FOP) is a rare congenital autosomal dominant disorder also involving HO (*Bravenboer et al., 2015*). Ectopic bones form progressively through endochondral ossification, mostly in episodic flare-ups associated with inflammation (*Towler and Shore, 2022*). These cumulative osteogenic events lead to reduced mobility resulting from ankylosing joints. Affected individuals have a shorter life span, most commonly due to thoracic insufficiency syndrome (*Kaplan et al., 2010*; *Pignolo et al., 2016*).

HO reflects a shift from a normal tissue repair process to an aberrant reactivation of bone-forming developmental programs that require acute inflammation and the excessive expansion of local progenitors, followed by inappropriate differentiation of these progenitors into chondroblasts and osteoblasts which finally results in bone formation (*Hwang et al., 2022*). In both trauma-induced HO and FOP, pathology appears following the excessive activation of receptors sensitive to transforming growth factor-β (TGF-β) superfamily members (*Lees-Shepard et al., 2018*; *Wang et al., 2018*). FOP arises from gain-of-function mutations in the bone morphogenetic protein (BMP) type I receptor, encoded by the *ACVR1* gene, with the most common mutation being c.617G>A, R206H (*Shore et al., 2006*). Higher SMAD1/5-mediated signaling of mutated ACVR1 has been partially attributed to a loss of auto-inhibition of the receptor and mild hypersensitivity to BMP ligands (*Chaikuad et al., 2012*; *Groppe et al., 2011*; *van Dinther et al., 2010*). Signaling by mutated ACVR1 still relies on ligand-induced heterotetrameric clustering but, unlike wild-type ACVR1, does not require ACVR2A/B kinase activity (*Agnew et al., 2021*; *Ramachandran et al., 2021*). More importantly, mutated ACVR1 receptors alter their signaling specificity, abnormally activating SMAD1/5 and altering the non-canonical BMP signals (e.g. p38 or PI3K) in response to activin A (*Hatsell et al., 2015*; *Hino et al., 2015*; *Valer et al., 2019*). Accordingly, evidence supports that activin A is necessary and sufficient for HO in FOP (*Hatsell et al., 2015*; *Lees-Shepard et al., 2018*; *Upadhyay et al., 2017*). In contrast, activin A does not drive post-traumatic HO (*Hwang et al., 2020*), and non-genetically driven HO mostly arises from excessive BMP and TGF-β signaling, with functional redundancy between different type I receptors of the TGF-β superfamily (*Agarwal et al., 2017*; *Patel et al., 2022*; *Sorkin et al., 2020*; *Wang et al., 2018*).

Evidence points to mesenchymal fibroadipogenic precursors (FAPs), which are widely distributed in muscle and other connective tissues, as the key cell-of-origin that aberrantly undergo chondrogenesis and further ectopic bone formation (*Dey et al., 2016*; *Eisner et al., 2020*; *Lees-Shepard et al., 2018*). However, both FOP and non-genetic HO also require local tissue destruction and inflammation, which indicate that FAPs should be invariably primed for ectopic bone formation by this inflammatory microenvironment (*Barruet et al., 2018*; *Convente et al., 2018*; *Hwang et al., 2022*; *Matsuo et al., 2019*; *Pignolo et al., 2013*; *Sorkin et al., 2020*).

Inflammation in HO is characterized by an initial acute response following injury with the activation of innate immunity and the influx of neutrophils and monocytes (*Convente et al., 2018*; *Hwang et al., 2022*). Additionally, B and T cells of the adaptive system are also recruited (*Chakkalakal et al., 2012*; *Convente et al., 2018*). However, mice lacking B or T lymphocytes exhibited no delay in the development of HO after injury, indicating that these cells may play a subsequent role in the dissemination of bone lesions (*Kan et al., 2009*). During intermediate and late inflammatory stages, the recruitment of monocytes, macrophages, and mast cells occurs, which in turn exerts autocrine and paracrine effects on nearby FAPs (*Chakkalakal et al., 2012*; *Convente et al., 2018*; *Sorkin et al., 2020*; *Tu et al., 2023*). Highlighting their relevance in HO, the depletion of mast cells and macrophages profoundly impairs genetic and non-genetic HO (*Convente et al., 2018*; *Torossian et al., 2017*; *Tu et al., 2023*). BMP receptors are robustly expressed in monocytes and macrophages and the expression of ACVR1[R206H] has been shown to extend inflammatory responses in patient-derived macrophages (*Barruet et al., 2018*; *Matsuo et al., 2021*; *Matsuo et al., 2019*). Among the plethora of cytokines secreted by monocytes, macrophages, and mast cells, both activin A and TGF-β stand out as extremely relevant for HO (*Alessi Wolken et al., 2018*; *Hatsell et al., 2015*; *Lees-Shepard et al., 2018*; *Patel et al., 2022*; *Sorkin et al., 2020*; *Upadhyay et al., 2017*).

Genetic and pharmacological studies have indicated that osteochondroprogenitor specification and maturation depend on phosphatidylinositol 3-kinase-α (PI3Kα) (*Ford-Hutchinson et al., 2007*; *Fujita et al., 2004*; *Ikegami et al., 2011*). We found that PI3K signaling was also linked to HO, since inhibitors of PI3Kα (BYL719/Alpelisib/Piqray) prevented HO in mouse models without major side-effects (*Valer et al., 2019*). Mechanistically, PI3Kα inhibitors hamper canonical and non-canonical BMP signaling, decreasing total and phosphorylated SMAD1/5 levels and reducing transcriptional responsiveness to BMPs/activin A in mesenchymal progenitors (*Gámez et al., 2016*; *Valer et al., 2019*). In the current study, we demonstrate that the pharmacological and genetic inhibition of PI3Kα in HO progenitors at injury sites reduces HO in vivo. Moreover, envisioning a future translation into the clinic, we have optimized the administration of BYL719 and found that the delayed administration of BYL719, up to 7 days after inflammatory injury, still prevents HO in mice. In addition, we found that BYL719 blocks osteochondroprogenitor specification and effectively reduces the essential inflammatory response. Altogether, the data presented here show the potent therapeutic effect of PI3Kα inhibition on HO in mouse models.

## Results

### Delayed initiation of treatment with PI3Kα inhibitor effectively prevents HO

HO takes place following a precise temporal pattern of progenitor activation and the recruitment of distinct cell types at the ossification centers, which causes early divergence from the normal skeletal muscle repair program. We previously found that the pharmacological administration of BYL719 prevents HO in a mouse model of HO (*Valer et al., 2019*). Given that BYL719 is marketed for the treatment of cancer and overgrowth syndrome, it is a promising therapeutic molecule in pathological HO. To further determine a potential therapeutic window for BYL719 and better understand the cellular targets of BYL719, we evaluated the effect of BYL719 when administered intermittently and several days after HO induction. For this, we used a conditional mouse model (*Acvr1$^{Q207Dfl/fl}$*) in which we express Cre recombinase through injection of CRE-expressing adenoviral particles and cardiotoxin intramuscularly in the hindlimb (*Fukuda et al., 2006*). In this model, we administered intermittently BYL719 (i.p. 25 mg/kg) or vehicle control starting 1, 3, or 7 days after HO induction. We also implemented a different administration regimen of BYL719 only for the initial 3 days following the injury (*Figure 1A*). None of the treatments led to significant changes in mouse body weight (*Figure 1—figure supplement 1*). HO volume was analyzed by micro computed tomography (μCT) 23 days post-injury. The early administration of the treatment with BYL719 at day 1 or 3 after the onset of HO resulted in significantly lower HO; late administration 7 days after the induction of HO was also partially effective (*Figure 1B, C*). Discontinuation of the treatment after 3 days did not prevent HO at day 23. These results indicate that the inhibition of PI3Kα by BYL719 after injury prevents HO, and this protection is still effective if the treatment is started several days after injury, expanding its therapeutic window.

### Deficiency of PI3Kα at injury sites is sufficient to partially prevent HO

Since other small molecules designed to target PI3Kα have reported partial off-target effects on kinases other than PI3Kα (*Furet et al., 2013*; *Jamieson et al., 2011*), we aimed to confirm that the results obtained with BYL719 were due to the specific inhibition of PI3Kα. Therefore, we developed a conditional mouse model (*Acvr1$^{Q207Dfl/fl}$:Pik3ca$^{fl/fl}$*) in which the Cre recombinase simultaneously drives the expression of ACVR1$^{Q207D}$ and the deletion of the catalytic subunit of PI3Kα (*Pik3ca*) upon intramuscular hindlimb combined injection of Adenovirus-Cre and cardiotoxin to trigger HO. With this approach, Cre mediates the expression of ACVR1$^{Q207D}$ and the deletion of PI3Kα in the same cells, whereas cells recruited afterwards will remain wild type for both genes. As expected, μCT analysis performed 23 days after injury showed extensive HO in mice expressing ACVR1$^{Q207D}$, which was prevented by intermittent treatment with BYL719 initiated 1 day after injury (*Figure 2A, B, D*). Genetic deletion of *Pik3ca* in mice expressing ACVR1$^{Q207D}$ led to a reduction in HO, which was further enhanced by BYL719 (*Figure 2B*). Importantly, no significant changes in the weight of the mice were observed, confirming the absence of severe toxicity (*Figure 2—figure supplement 1A*). Histomorphometrically, ACVR1$^{Q207D}$ mice wild type for PI3Kα developed islands and/or bone spurs with a

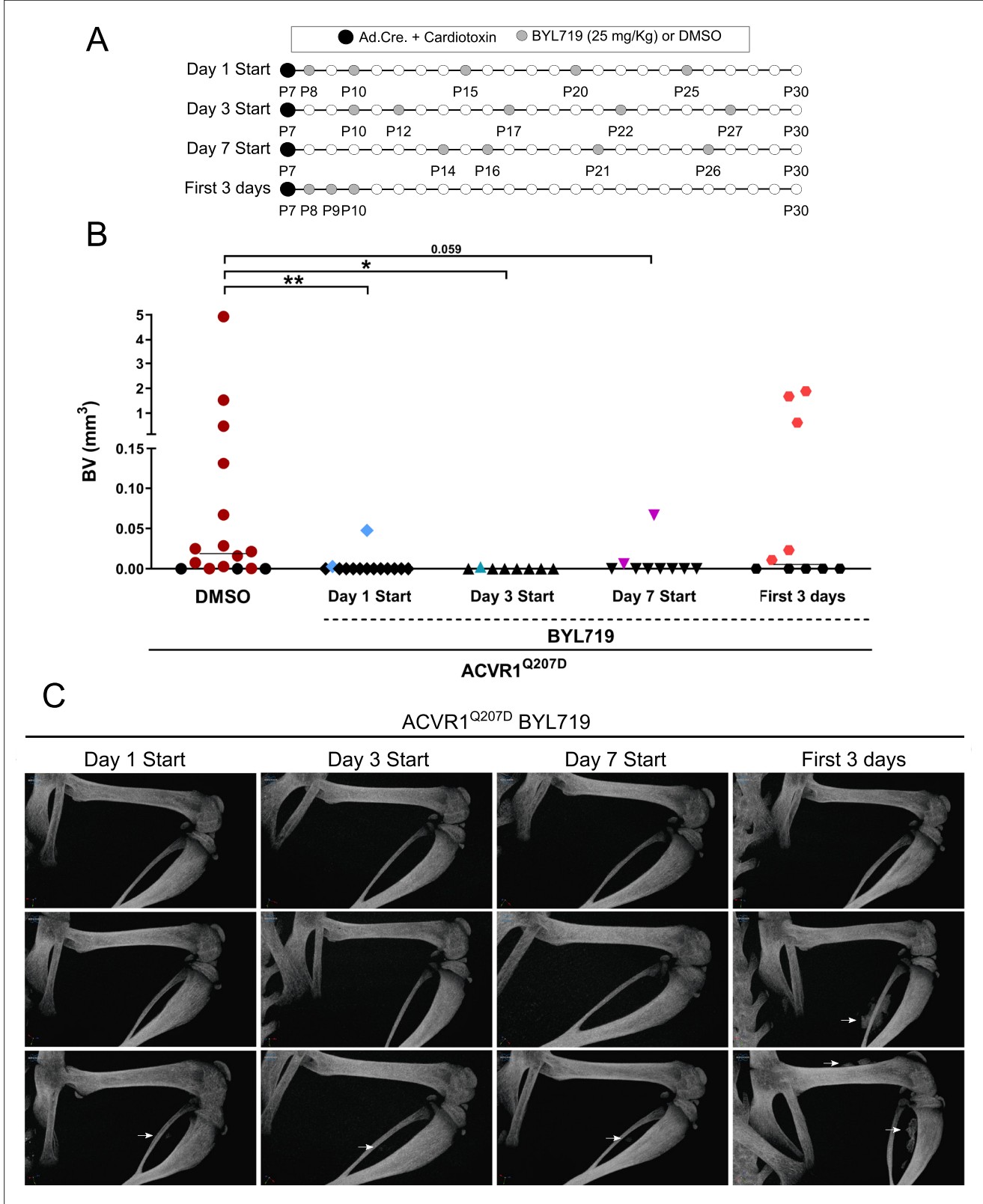

**Figure 1.** Delayed initiation of treatment with PI3Kα inhibitor effectively prevents heterotopic ossification. (**A**) Heterotopic ossification was induced at P7 through the injection of Adenovirus-Cre (Ad.Cre) and cardiotoxin in the mice hindlimb. The pattern of five administrations of DMSO or BYL719 (25 mg/kg) is indicated by gray dots started at P8 (start day 1), P10 (delayed start day 3), and P14 (delayed start day 7). One group was administered DMSO or BYL719 (25 mg/kg) only at P8, P9, and P10 (first 3 days post-HO-induction), as indicated by gray dots. The final time point, P30, was conserved between

*Figure 1 continued on next page*

*Figure 1 continued*

experimental groups. (**B**) Quantification of heterotopic ossifications bone volume (BV) (mm³) of each experimental group. Colored symbols indicate the presence of heterotopic ossifications. Black symbols indicate the absence of heterotopic ossifications. Individual mouse values with group median are shown. *p < 0.05, **p < 0.01, Kruskal–Wallis test with Dunn's multiple comparisons test. (**C**) Representative 3D microtomography images of the injected hindlimbs of three different mice for each experimental group. White arrows point to heterotopic ossification.

The online version of this article includes the following figure supplement(s) for figure 1:

**Figure supplement 1.** Mice body weight from P8 (1 day post-HO-induction) to P30.

well-organized bone structure. However, in PI3Kα deficient mice, ACVR1$^{Q207D}$ expression only led to minor ectopic calcifications that were already surrounded by fully regenerated muscle tissue on the 23rd day after injury (*Figure 2D*, *Figure 2—figure supplement 1B*, *Figure 2—figure supplement 2*). Accordingly, analysis of the ratio bone volume/tissue volume (BV/TV) and the correlation between bone volume and BV/TV of individual mice was clearly different between *Pi3kca* genotypes, irrespective of BYL719 treatment (*Figure 2C*, *Figure 2—figure supplement 1C*). These results demonstrate that the abrogation of PI3Kα activity (either genetically or pharmacologically) in cells at injury sites is relevant for their inhibitory effects in HO. This might be attributed to the direct effects of BYL719 on progenitor cell expansion and chondroblast specification, as well as its possible ability to reduce the inflammatory response required for HO.

## Genetic or pharmacological inhibition of PI3Kα prevents osteochondroprogenitor specification and reduces the number of FAPs at injury sites

Next, we aimed to assess the effects of PI3Kα inhibition in osteochondroprogenitor cells. For this, we isolated bone marrow-derived mesenchymal stem cells (BM-MSCs) from *Pik3ca*$^{fl/fl}$ mice and transduced them ex vivo with retrovirus expressing *Acvr1* (*wild type* and *R206H*) with or without Cre recombinase. The gene expression levels of exogenous mRNA of *Acvr1* and *Acvr1*$^{R206H}$ was similar (*Figure 3A*). As anticipated, transduction with Cre viruses significantly reduced the expression levels of *Pik3ca* to approximately 40%. This is consistent with an estimated transduction efficiency of nearly 60% (*Figure 3B*). The overexpression of *Acvr1*$^{R206H}$ increased basal and activin-dependent expression of canonical (*Id1* and *Sp7*) and activin-dependent expression of non-canonical (*Ptgs2*) BMP target genes (*Figure 3C*), known markers and drivers of osteochondrogenic differentiation (*Nakashima et al., 2002*; *Wang et al., 2013*). Both pharmacological and genetic inhibition of PI3Kα reduced canonical and non-canonical ACVR1$^{R206H}$ transcriptional responses induced by activin A. Of note, stimulation with activin A increased its own transcription (*Inhba*) in *Acvr1*$^{R206H}$-transduced BM-MSCs, which was also inhibited by BYL719 (*Figure 3C*). Therefore, BYL719 may also prevent osteochondroprogenitor differentiation indirectly, via transcriptional inhibition of *INHBA*/activin A expression. These results demonstrate a direct effect of BYL719 on osteochondroprogenitor cells and suggest that the effects of BYL719 on ACVR1-downstream signaling are mainly on target, due to PI3Kα kinase inhibition.

In addition, we analyzed the number of FAPs during the progression of ectopic bone formation in our in vivo mouse model of HO. Samples of injured muscles of the conditional mouse model (*Acvr1*$^{Q207Dfl/fl}$) were subjected to immunofluorescence to identify the number of PDGFRA+ cells (FAPs) at 4, 9, 16, and 23 days after induction of HO. The number of FAPs remained sustained until day 16th after injury, decreasing afterwards (*Figure 3—figure supplement 1A, B*). More importantly, treatment with BYL719 reduced the number of PDGFRA+ cells throughout the ossification process. We also observed an increase in the diameter of myofibers in animals treated with BYL719 (*Figure 3—figure supplement 1A, C*). These results suggest improved muscle regeneration in animals treated with BYL719 compared to those undergoing HO.

It is known that 2-aminothiazole-derived PI3Kα inhibitors could also inhibit ACVR1 kinase activity (*Jamieson et al., 2011*). Therefore, we investigated whether BYL719 effects could be explained by the reduction in ACVR1 kinase activity. For this, we made use of a kinase target engagement approach. Upon transient transfection of Nanoluciferase-tagged TGF-β receptors, we incubated the cells with BYL719 and/or an ATP-like tracer analog. In the absence of an ATP competitor molecule, the tracer analog and Nanoluciferase enzyme stay in close proximity, thereby allowing for bioluminescence resonance energy transfer (BRET) between the Nanoluciferase (energy donor, emitting at 460 nm) and the

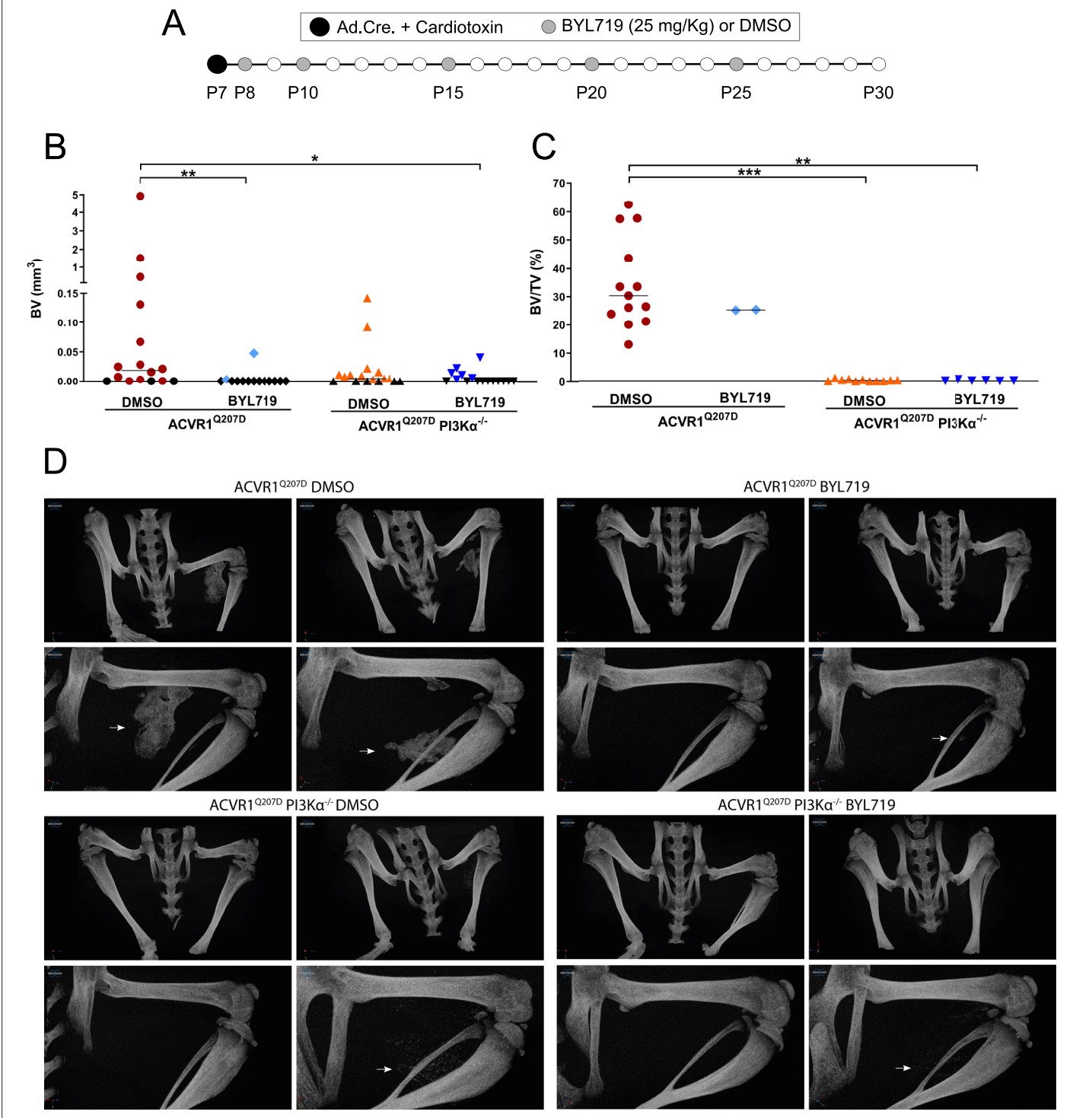

**Figure 2.** Deficiency of PI3Kα at injury sites is sufficient to partially prevent heterotopic ossification. (**A**) Heterotopic ossification (HO) was induced at P7 through the injection of Adenovirus-Cre (Ad.Cre) and cardiotoxin in the mice hindlimb. Either DMSO (vehicle) or BYL719 (25 mg/kg) was injected following the scheme indicated with gray dots, starting at P8 (**B**). Quantification of HOs bone volume (BV) (mm³) of each experimental group. Colored symbols indicate the presence of HOs. Black symbols indicate the absence of HOs. Individual mouse values with group median are shown. *p < 0.05, **p < 0.01, Kruskal–Wallis test with Dunn's multiple comparisons test. (**C**) Quantification of the ratio of bone volume per tissue volume (bone volume/tissue volume, BV/TV) within HOs of each experimental group, including only mice with detected HO. Data shown are of each individual mouse with the group median. **p < 0.01, ***p < 0.001, Kruskal–Wallis test with Dunn's multiple comparisons test. (**D**) Representative 3D frontal microtomography

*Figure 2 continued on next page*

*Figure 2 continued*

images of the injected hindlimbs of mice for each experimental group and a detailed close-up image for each selected mouse. White arrows show HO in the close-up images.

The online version of this article includes the following figure supplement(s) for figure 2:

**Figure supplement 1.** Absence of toxic effects of PIk3ca deletion and BYL719 administration.

**Figure supplement 2.** Muscle sections of mice 23 days after injury stained with Masson's Trichrome.

tracer analog (energy acceptor) Once excited, the acceptor molecule emits fluorescence (610 nm) (*Figure 4A, B*). BYL719 incubated at 1 and 10 µM failed to significantly inhibit the acceptor emission of the BMP type I receptors ACVRL11, ACVR1, BMPR1A, ACVR1B, TGFBR1, and the mutant receptor ACVR1$^{R206H}$ (*Figure 4C*), without significant changes in the protein levels of each receptor (*Figure 4—figure supplement 1*). In addition, we demonstrated that BYL719 does not target the activity of the type II kinase receptors ACVR2A, ACVR2B, BMPR2, and TGFBR2. We confirmed these results using in vitro kinase activity assays with recombinant ACVR1$^{R206H}$. BYL719 exhibited no major effects on ACVR1$^{R206H}$ activity up to 10 µM (*Figure 4D*).

Given that BYL719 does not directly target the activity of ACVR1 or related receptors, we established two in vitro models to further profile the main targeted pathways. In human MSCs (hMSCs) stably overexpressing ACVR1 or ACVR1$^{R206H}$, recombinant activin A was able to induce SMAD1/5 activation and the expression of ACVR1-downstream targets *ID1* and *ID3* in cells overexpressing ACVR1$^{R206H}$ (*Figure 5—figure supplement 1A, B*). We performed micromass chondrogenic differentiation assays in differentiation media supplemented with TGFβ1, with or without activin A and BYL719 in parental cells and cells transduced with ACVR1$^{WT}$ or ACVR1$^{R206H}$. Coincubation with BYL719 (10 µM) completely inhibited the formation of a chondrogenic glycosaminoglycans-rich matrix in response to TGFβ1 plus activin A in the different cell types (*Figure 5A, B*) and partially abrogated the expression of the chondrogenic genes *ACAN* and *MMP13* (*Figure 5C*). Furthermore, in these cells, coincubation with BYL719 (10 µM) was able to repress activin A-induced alkaline phosphatase activity upon 7 and 11 days of culture, respectively (*Figure 5D, E*). In addition, to confirm these results, we isolated MSCs from UBC-CRE-ERT2/ACVR1$^{R206H\ fl/wt}$ knock-in mice. In these mice, the conditional ACVR1$^{R206H}$ construct was knocked-in to the endogenous *Acvr1* gene immediately following intron 4. After 4-OH tamoxifen addition, CRE activity excises murine *Acvr1* exons 5–10 and induce expression of the corresponding exons of human ACVR1$^{R206H}$ and an eGFP marker. Therefore, these cells, when treated with 4OH tamoxifen, express the intracellular exons of human *ACVR1$^{R206H}$* in the murine *Acvr1* locus. ACVR1$^{R206H}$ mutant cells display an enhanced chondrogenic response to activin A compared to wild-type cells (*Figure 5F*). In both wild type and mutant MSCs, treatment with BYL719 decreased the expression of chondrogenic. These results confirmed that BYL719 could inhibit chondrogenic differentiation of human and murine MSCs irrespective of the expression of either wild type or mutant ACVR1.

To further identify the effects of BYL719, we performed bulk RNA sequencing (GSE237512) where we preincubated hMSCs-ACVR1$^{WT}$ or hMSCs-ACVR1$^{R206H}$ with or without BYL719 for 30 min, which was followed by stimulation with two high-affinity ligands for ACVR1, activin A (50 ng/ml), or BMP6 (50 ng/ml) for 1 hr (*Figure 6A*). Consistent with our results on osteogenic progenitor cells, gene ontology analysis of differentially expressed genes between cells expressing ACVR1$^{WT}$ and ACVR1$^{R206H}$ resulted in the robust regulation of relevant biological processes. Ossification (GO:0001503) and osteoblast differentiation (GO:0001649) were detected as two of the ten most significantly differentially regulated biological processes between these conditions (*Figure 6B*). Moreover, using gene ontology, we analyzed ossification and osteoblast differentiation biological processes in the presence of ACVR1$^{WT}$ or ACVR1$^{R206H}$ receptor, with different ligands (BMP6 or activin A), and with or without BYL719 inhibitor. The addition of BYL719 (1 µM) resulted in a downregulation of these GO terms for all conditions except activin A stimulated ACVR1$^{WT}$ cells (*Figure 6C*).

When comparing gene expression profiles between ACVR1$^{R206H}$ treated with activin A, with or without BYL719, gene set enrichment analysis (GSEA) of our GO terms of interest showed negative normalized enrichment scores, consistent with our gene ontology analysis (*Figure 6D*). By looking at the leading edge, we could investigate the most relevant genes within these GO terms, which included relevant genes of these pathways such as *PTGS2*, *MSX2*, *SOX9*, and *BMP2* (*Figure 6E*). Moreover, several inflammatory cytokine genes (e.g. *IL6* and *IL15*) and inflammatory signaling pathways (e.g.

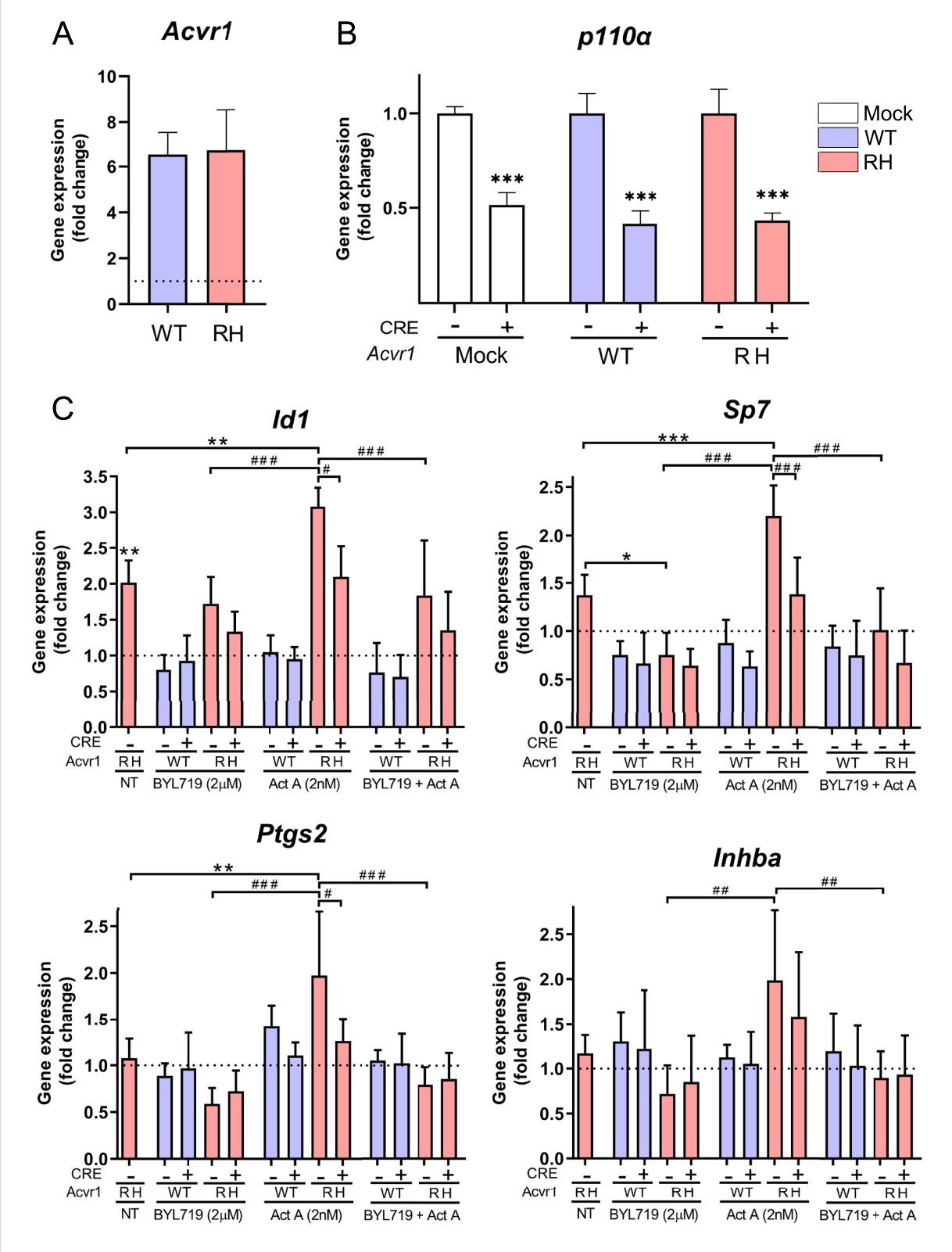

**Figure 3.** Analysis of *Acvr1* gene expression by qPCR in bone marrow-derived mesenchymal stem cells (BM-MSCs) from *Pik3ca*^fl/fl mice infected with virus expressing wild-type *Acvr1* (WT) or *Acvr1*^R206H^ (RH). (**A**) The endogenous expression level of the *Acvr1* gene in mock-transfected BM-MSCs is shown as a dotted horizontal line. Data are shown as mean ± SD (*n* = 12 per group). Unpaired *t*-test between transfected groups. (**B**) Gene expression analysis of *Pik3ca*α in BM-MSCs from *Pik3ca*^fl/fl mice, infected with virus expressing wild-type *Acvr1* (WT) or *Acvr1*^R206H^ (RH) and/or Cre co-infection. Data are

*Figure 3 continued on next page*

*Figure 3 continued*

shown as mean ± SD (*n* = 3 per group). ***p < 0.001, two-way ANOVA with Tukey's multiple comparisons test. (**C**) mRNA expression of canonical (*Id1* and *Sp7*), non-canonical (*Ptgs2*) target genes, and activin A (*Inhba*) in BM-MSCs *Pik3ca*^fl/fl transfected with *Acvr1* (wild type or R206H) with or without Cre recombinase. Cells were NT (not treated) or treated with BYL719 (2 μM) and/or activin A (2 nM). Expression data were normalized to those of control cells which were transfected only with *Acvr1* WT without any treatment, shown as a dotted horizontal line. Asterisks (*) refer to the differences between different conditions of *Acvr1* RH cells compared to control cells. Hash signs (#) refer to the differences between different conditions of *Acvr1* RH cells compared to *Acvr1* RH cells without Cre recombinase and treated with activin A. Data are shown as mean ± SD (*n* = 6 per group). * or # p < 0.05, ** or ## p < 0.01, *** or ### p < 0.001, two-way ANOVA with Tukey's multiple comparisons test.

The online version of this article includes the following figure supplement(s) for figure 3:

**Figure supplement 1.** BYL719 reduces the number of FAPs and improves muscle regeneration during the ossification process.

TNF and NF-κB), were downregulated by BYL719 (*Figure 6—figure supplement 1A, B*). Therefore, we hypothesized that, in addition to preventing the activation of osteogenic and chondrogenic differentiation pathways in progenitor cells, part of the mechanism by which BYL719 prevents HO in vivo may be due to the modulation of inflammatory responses.

## PI3Kα inhibition reduces the hyper-inflammatory response in HO

To confirm whether BYL719 can modulate excessive inflammation at the injured sites, we examined its impact on monocytes, macrophages, and mast cells in vivo. These cell populations are known to increase in number 2–4 days after injury (*Convente et al., 2018*; *Sorkin et al., 2020*), which coincides with the effects observed following early administration of BYL719. Hindlimb muscle samples were collected at 2, 4, 9, 16, and 23 days post-induction of HO with Cre viruses and cardiotoxin in *Acvr1*^Q207Dfl/fl mice (*Figure 7A*). The group of treated animals received BYL719 intermittent treatment starting 1 day after injury. After 16 days, structures of mineralized bone became detectable in vehicle-treated mice (*Figure 7—figure supplement 1A, B*). The number of F4/80-positive monocytes/macrophages was high 2 days after injury and remained elevated in the HO lesions throughout the ossification process (*Figure 7B, C*). BYL719-treated animals showed an almost 50% reduction in the number of F4/80-positive cells at day 2 post-injury onwards. Consistent with previous studies, we observed an increased number of mast cells at injury sites. Their numbers peaked at 4 days and only dropped 23 days after injury in vehicle-treated mice (*Figure 7D, E*). In BYL719-treated mice, mast cells were increased up to day 9, but their number was completely normalized to preinjury levels already at day 16 (*Figure 7D, E*). This observation is consistent with the complete muscle regeneration observed in BYL719-treated mice compared to untreated mice, which are actively developing ossifications at that time (*Figure 7—figure supplement 2*). These results confirm a functionally relevant dual role of PI3Kα inhibition, by preventing the differentiation of osteochondrogenic progenitor cells and also by reducing the local inflammatory response.

## PI3Kα inhibition reduces proliferation, migration, and inflammatory cytokine expression in monocytes, macrophages, and mast cells

Next, we used several models of disease-relevant immune cells to study the effect of pharmacological PI3Kα inhibition. Given the technical difficulties in transducing immune cells with lentiviral particles carrying ACVR1 R206H, we decided to partially recapitulate ACVR1 R206H activation with recombinant BMP6 and to test the effect of BYL719 in these conditions. Activation of ACVR1 signaling with BMP6 did not significantly modify the proliferation rate of any of the cell lines tested, that is, human monocytes (THP1), murine macrophages (RAW264.7) and human mast cells (HMC1) (*Figure 8A–C*). However, incubation with BYL719 in vitro strongly reduced their growth rate at 2 μM and almost completely arrested their proliferation at 10 μM. It is well known that monocytes are actively recruited from circulation to the ossification sites early after injury (*Sorkin et al., 2020*). We also examined the effects of BYL719 in proliferation in additional cell types involved in HO, such as myoblasts and mesenchymal progenitors. BYL719 reduced the proliferation of myoblast and mesenchymal progenitors in vitro (*Figure 8—figure supplement 1A, B*). However, the reduction in the proliferation did not reach the extent observed in monocytes or macrophages. We expanded our analysis by interrogating chemotactic migration of monocytes in transwell assays. Whereas monocytes actively migrated toward FBS as a chemotactic agent, the incubation of monocytes with BYL719 deeply diminished their

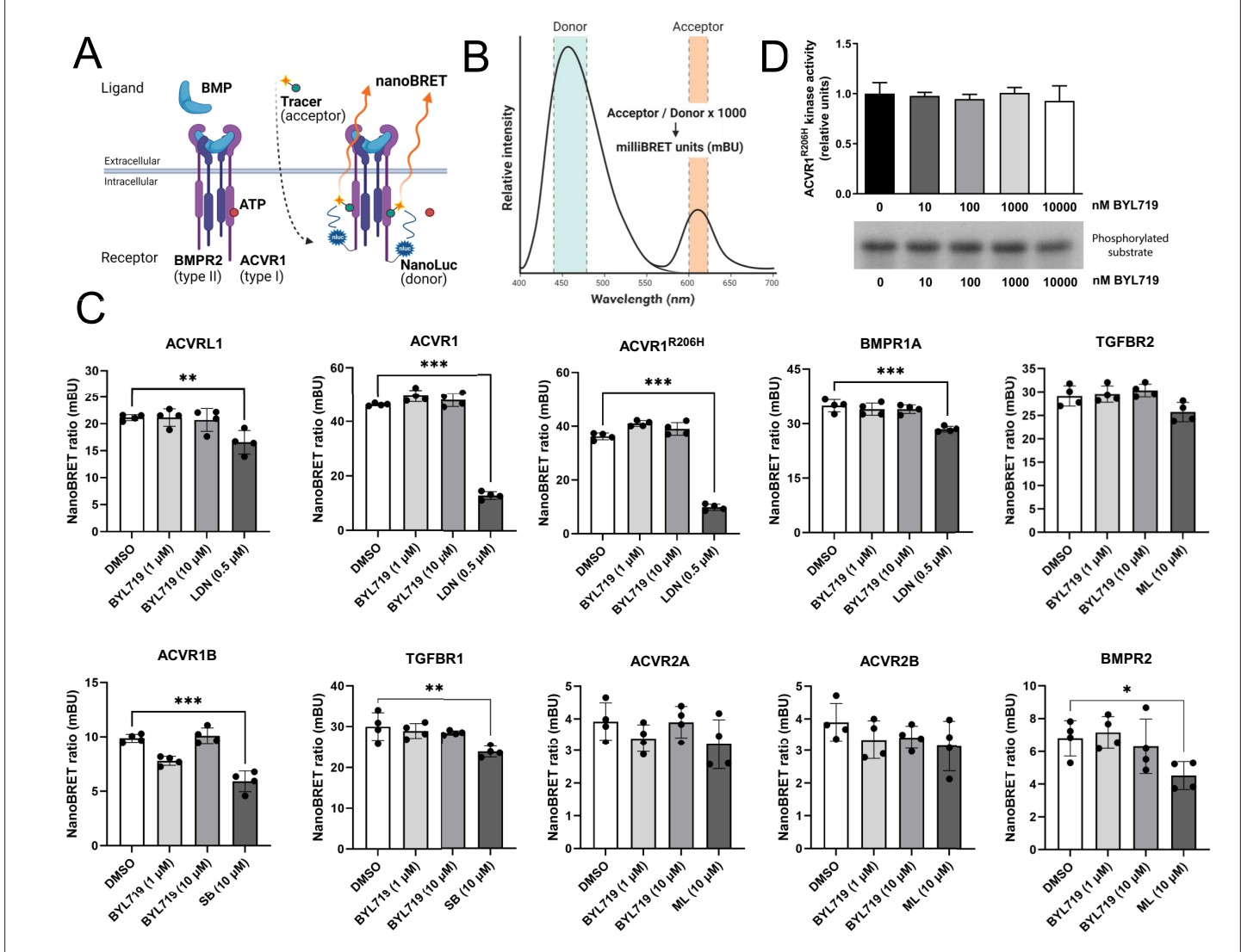

**Figure 4.** BYL719 does not reduce ACVR1 kinase activity. (**A**) A schematic depiction of nano-bioluminescence resonance energy transfer (nanoBRET) target engagement assays. TGF-β receptors–Nanoluciferase fusion proteins are expressed in COS-1 cells and an ATP-like tracer analog in close proximity with the Nanoluciferase donor allows energy transfer from the Nanoluciferase donor to the fluorescent tracer acceptor. An ATP analog molecule (type I inhibitor) will compete with the fluorescent tracer, impairing close proximity donor–acceptor and reducing the nanoBRET ratio. (**B**) The nanoBRET emission spectra consist of the Nanoluciferase donor (460 nm) and the fluorescent acceptor (610 nm). The nanoBRET ratio is shown as milliBRET units (mBU) by dividing the acceptor emission by the donor emission times 1000. (**C**) NanoBRET target engagement analyses of ACVRL1, ACVR1, ACVR1$^{R206H}$, BMPR1A, ACVR1B, TGFBR1, TGFBR2, ACVR2A, ACVR2B, and BMPR2 testing 1 or 10 µM BYL719 with $n = 4$. As controls, LDN193189 (0,5 µM), SB431542 (10 µM), and ML347 (10 µM) were used. Data are shown as mean ± SD. One-way ANOVA with Dunnett's multiple comparisons test. (**D**) Casein phosphorylation by ACVR1$^{R206H}$ kinase. Phosphorylation was performed in the presence of ACVR1$^{R206H}$ kinase and increasing concentrations of the PI3Kα inhibitor BYL719. Quantification of kinase activity. Data are shown as mean ± SD ($n = 4$ independent experiments). One-way ANOVA with Tukey's multiple comparisons test. *$p < 0.05$, **$p < 0.01$, ***$p < 0.001$.

The online version of this article includes the following source data and figure supplement(s) for figure 4:

**Source data 1.** Original files for autoradiographies displayed in *Figure 4D*.

**Source data 2.** PDF file containing original autoradiographies for *Figure 4D*, indicating the relevant bands and treatments.

**Figure supplement 1.** Quantification of the TGF-β receptor–Nanoluciferase protein expression levels measured by the raw donor emission (excitation at 450–480 nm).

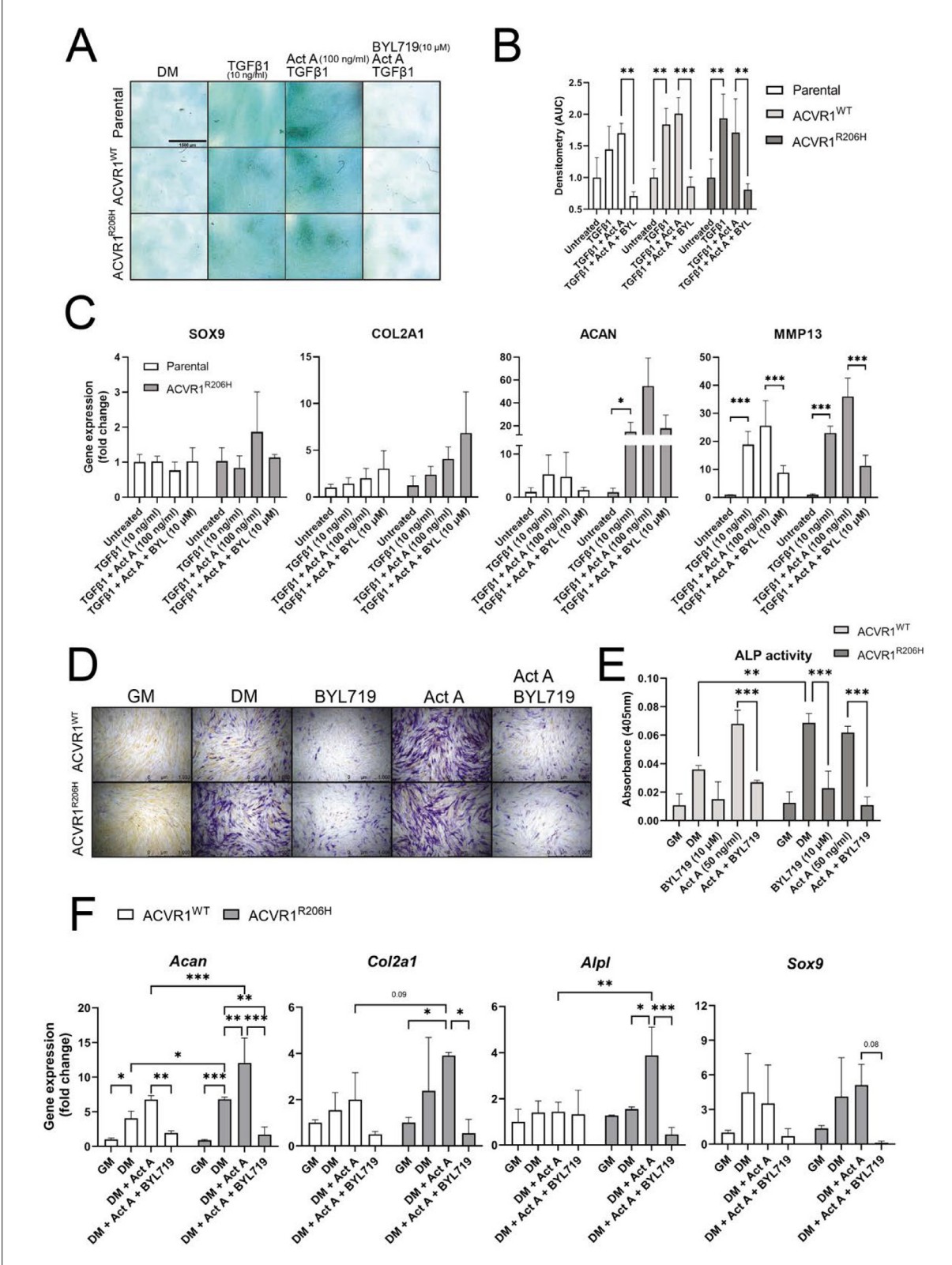

**Figure 5.** Analyses of chondrogenic progenitor specification of parental human MSCs (hMSCs), hMSC-ACVR1<sup>WT</sup> or hMSC-ACVR1<sup>R206H</sup> treated with TGF-β1 (10 ng/ml) and activin A (100 ng/ml), with and without BYL719 (10 μM) as indicated (**A**). Representative images of Alcian Blue staining of micromass cultures after 3 weeks of chondrogenic differentiation. Scale bar represents 1500 μm. (**B**) Densitometry analysis of the Alcian Blue staining in (**A**) (*n* = 3 per group). (**C**) RT-qPCR results of *SOX9*, *COL2A1*, *ACAN*, and *MMP13* after chondrogenic progenitor specification (*n* = 4 per condition).

*Figure 5 continued on next page*

*Figure 5 continued*

(D) Representative images of the alkaline phosphatase (ALP) staining after 1 week of chondrogenic progenitor specification. Scale bar represents 1000 µm. (E) Quantification of the ALP activity (n = 3 per group). (F) RT-qPCR results of *Acan*, *Col2a1*, *Alpl*, and *Sox9* after chondrogenic progenitor specification from *Acvr1*[WT] or ACVR1[R206H] MSCs treated with activin A (100 ng/ml), with or without BYL719 (10 µM) in TGFβ1-containing differentiation media (DM) as indicated (n = 3 per condition). Data are shown as mean ± SD. In all figure panels, significance is detailed between the indicated conditions. *p < 0.05, **p < 0.01, ***p < 0.001, two-way ANOVA with Tukey's multiple comparisons test.

The online version of this article includes the following source data and figure supplement(s) for figure 5:

**Figure supplement 1.** Human MSC-ACVR1[R206H] are responsive upon activin A (50 ng/ml) stimulation compared to hMSC-ACVR1[WT] after overnight starvation.

**Figure supplement 1—source data 1.** Original files for western blot analysis displayed in *Figure 5—figure supplement 1*.

**Figure supplement 1—source data 2.** PDF file containing original western blots for *Figure 5—figure supplement 1*, indicating the relevant bands and treatments.

migratory ability (*Figure 8D*). These results suggest that BYL719 is able to diminish monocyte recruitment, while also reducing the expansion of inflammatory cells at injury sites.

Upon tissue injury, a multitude of pro- and anti-inflammatory cytokines are released at the site of injury. This triggers the recruitment and infiltration of various cell types, including immune cells and osteochondrogenic progenitors. We examined the expression of distinct cytokines upon ACVR1 stimulation with BMP6 in the presence of BYL719 in monocytic, macrophagic, and mast cell lines. In monocytes, BYL719 inhibited the expression of the pro-inflammatory cytokine *TNFA* and the expression of activin A (*INHBA*), whereas the expression of transforming growth factor *TGFB1* was increased by BMP6 and not modified by the addition of BYL719 (*Figure 8E*). *CCL2* (MCP-1), a chemokine inducing the recruitment of monocytes and macrophages into inflamed tissues as well as the M2 polarization of macrophages (*Sierra-Filardi et al., 2014*), was significantly induced by BMP6 and partially reduced by BYL719 (*Figure 8E*). In macrophages, BYL719 reduced the expression of pan-macrophage markers *Adgre1* (F4/80), *Cd68*, and the M1 macrophage marker *Cd80* (*Figure 8F*). Conversely, BYL719 increased the expression of the M2 macrophage marker *Il10*, regardless of the presence or absence of BMP6. In the human mast cell line HMC1, BYL719 inhibited the expression of the pro-inflammatory cytokine *IL6*, while the expression of *TGFB1*, *TNFA*, and activin A (*INHBA*) remained unchanged upon the addition of either BMP6 or BYL719 (*Figure 8G*). We also analyzed the effects of BYL719 on myogenic differentiation in vitro. Addition of 2 or 10 µM BYL719 was able to abolish the expression of the myogenic markers *Myod1* and myosin heavy-chain 1 (*Myh1*) while barely altering the expression of the myofibroblast marker α-SMA (*Acta2*) (*Figure 8—figure supplement 1C*). Similarly, BYL719 (at 2 or 10 µM) was able to block the formation of myotubes after 7 days of myogenic differentiation in vitro (*Figure 8—figure supplement 1D*). Altogether, these data suggest that PI3Kα inhibition might reduce the expression of certain pro-inflammatory cytokines. Moreover, even though these results should be further validated in cells carrying mutated ACVR1, the reduction in activin A expression by BYL719 observed in inflammatory cells and mesenchymal progenitors could be relevant for HO in FOP.

## Discussion

We previously showed that pharmacological PI3Kα inhibition using BYL719 has the potential to suppress HO by increasing SMAD1/5 degradation, reducing transcriptional responsiveness to BMPs, and blocking non-canonical responses such as the activation of AKT/mTOR (*Valer et al., 2019*). In this manuscript, we confirmed these results through both pharmacological and genetic approaches and further expanded the insights into the molecular and cellular mechanisms responsible for the therapeutic effect of PI3K inhibition. Here, we optimized the therapeutic window for BYL719 administration and found that the delayed administration of BYL719 effectively prevents HO. Moreover, in vivo simultaneous genetic inhibition of *Pik3ca* and the activation of mutated *Acvr1* at injury sites partially prevents HO, and we demonstrate that inhibition of PI3Kα inhibits osteochondroprogenitor specification. We determined that BYL719 does not inhibit recombinant ACVR1[R206H] kinase activity in vitro and does not behave as an ATP competitor inhibitor for any of the TGF-β receptor kinases. Finally, the administration of BYL719 prevented an exacerbated inflammatory response in vivo, possibly due to the effects observed on immune cell populations.

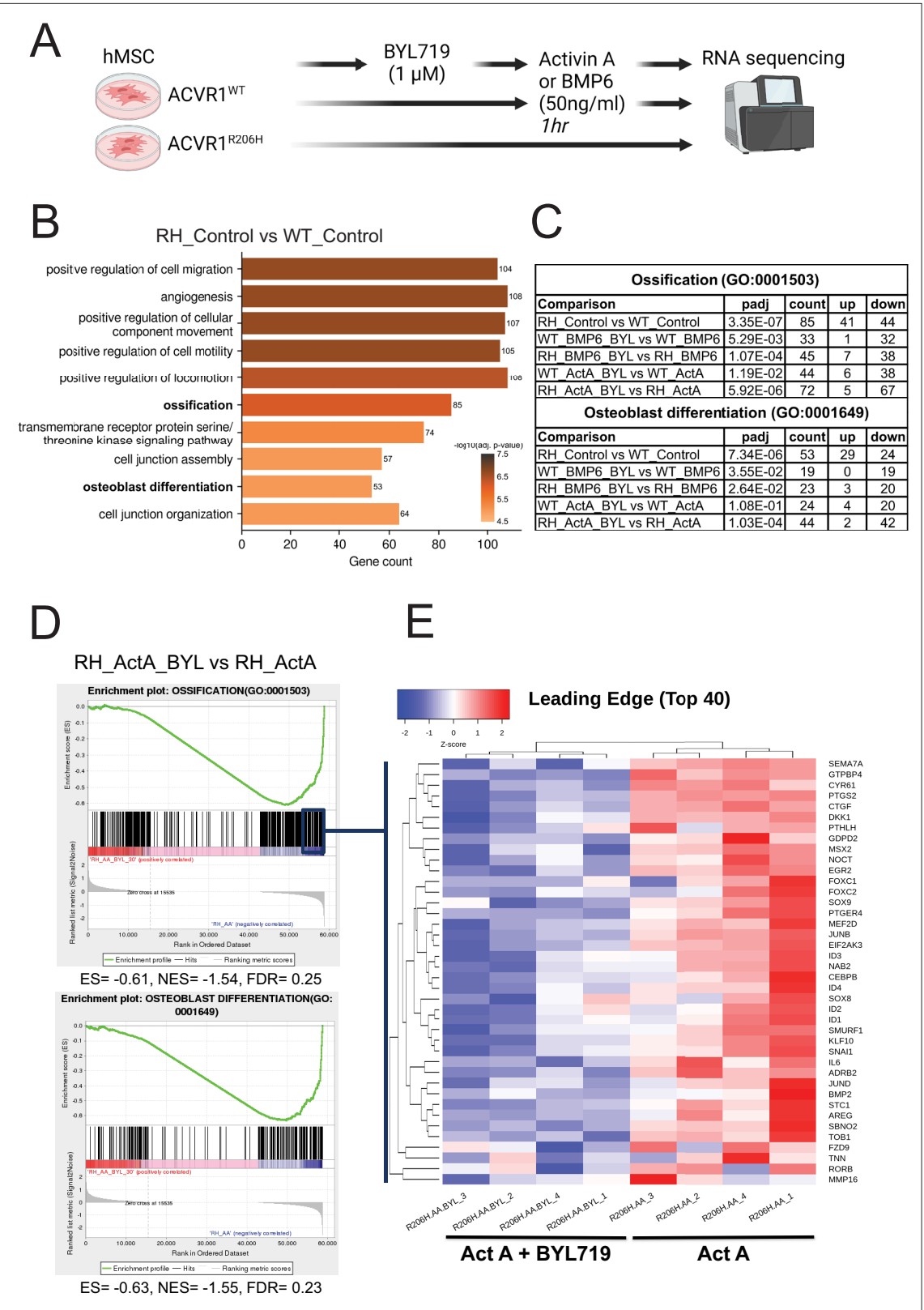

**Figure 6.** BYL 719 inhibits ossification and osteoblast differentiation processes in human ACVR1-R206H MSCs. (**A**) Schematic depiction of the experimental setup, involving bulk RNA sequencing of human MSC-ACVR1WT and ACVR1R206H upon overnight starvation, 30-min pre-treatment with or without 1 μM BYL719 and with or without 1 hr activin A (50 ng/ml) or BMP6 (50 ng/ml) stimulation. (**B**) The top 10 most significant gene ontology (GO) terms using all (up- and downregulated) differentially expressed genes between cells expressing ACVR1-WT and ACVR1-R206H under control conditions.

*Figure 6 continued on next page*

*Figure 6 continued*

Ossification (GO:0001503) and osteoblast differentiation (GO:0001649) were detected within the top 10 of the differentially regulated biological processes. (**C**) Table of GO terms ossification and osteoblast differentiation upon GO enrichment analysis in all tested conditions ACVR1[WT] or ACVR1[R206H] cells, stimulated with BMP6 or activin A and treated with BYL719. Statistical significance upon GO enrichment analysis, count of total differentially expressed genes (DEGs), and their classification as up- or downregulated DEGs are detailed for each comparison. (**D**) Gene set enrichment analysis (GSEA) with the groups RH_ActA_BYL vs RH_ActA, showing the enrichment plots of the gene ontology sets ossification (GO: 0001503) and osteoblast differentiation (GO:0001649). (**E**) A heatmap of the top 40 most relevant genes within the ossification GO geneset derived from the leading-edge subset of the enrichment plot (*n* = 4 per group). Sample names are detailed as receptor (R206H) ligand (AA, Act A, Activin A)_inhibitor (BYL719, if present)_ replicate#.

The online version of this article includes the following figure supplement(s) for figure 6:

**Figure supplement 1.** Gene set enrichment analysis (GSEA) of bulk RNA sequencing comparing hMSC-ACVR1[R206H] cells treated with BYL719 (1 μM) with untreated controls, both stimulated with activin A (50 ng/ml) (*n* = 4 per group).

While FAPs play a key role in bone formation in different types of HO, inflammation is also closely associated with all types of HO (*Barruet et al., 2018*; *Matsuo et al., 2019*). For instance, the depletion of mast cells and macrophages has been shown to reduce bone volume in FOP mouse models (*Convente et al., 2018*). Moreover, activin A, TGF-β, and other cytokines secreted by FAPs, monocytes, macrophages, and mast cells are essential for FOP and non-genetic HO (*Alessi Wolken et al., 2018*; *Hatsell et al., 2015*; *Lees-Shepard et al., 2018*; *Patel et al., 2022*; *Sorkin et al., 2020*; *Upadhyay et al., 2017*). In addition, ACVR1 antibodies can activate ACVR1[R206H] even in the absence of activin A, but muscular trauma is still required to induce HO (*Aykul et al., 2022*; *Lees-Shepard et al., 2022*), indicating the involvement of additional immune factors. Our study finds that BYL719, a PI3Kα inhibitor, reduces the expression of certain pro-inflammatory cytokines, effectively blocks the proliferation of monocytes, macrophages, and mast cells and reduces the migratory potential of monocytes. This likely contributes to a decreased number of monocytes and macrophages at injury sites and throughout the in vivo ossification process. It is known that excessive pro-inflammatory cytokine expression, including activin A, by monocytes and macrophages is observed in all types of HO and is induced by ACVR1[R206H] (*Matsuo et al., 2021*). Non-canonical signaling pathways (mTOR, p38, TAK1, and NF-κB) are also affected by ACVR1[R206H] in immune cell types (*Barruet et al., 2018*; *Hwang et al., 2022*). M1-type macrophages initiate acute inflammatory responses but later transition to M2-type macrophages associated with anti-inflammatory and reparative functions. Therefore, modulating macrophage phenotype toward regeneration may dampen HO (*Sorkin et al., 2020*). BYL719 reduces monocyte/macrophage numbers, migration, and pro-inflammatory cytokine expression, potentially altering their polarization. It also reduces activin A expression and promotes a shift toward the M2 phenotype. BYL719 can help to reduce the hyper-inflammatory state in HO, inhibit FAP expansion, and favor myogenic regeneration of muscle tissue (*Stanley et al., 2022*).

Our observation that the delayed administration of BYL719 is still effective has pathophysiological and therapeutic derivatives. Muscle injury in mice induces sequential changes in HO progression: early immune cell infiltration (days 1–3), a fibroproliferative and late inflammatory phase, with an expansion of FAPs and a spectrum of monocytes, macrophages, and mast cells as highly secretory cells (days 3–7), followed by chondrogenesis (days 7–14) and osteogenesis (days 14–23) (*Hwang et al., 2022*). Starting BYL719 administration 3 days post-injury is still fully therapeutically effective and even partially at 7 days post-injury. In addition, whereas untreated mice developed cartilage and bony lesions by day 9 and 16 post-injury, respectively, BYL719-treated animals fully regenerated muscle by day 16 post-injury. Altogether, this evidence suggests that the major effects of BYL719 occur between 3 and 16 days post-injury. We found that BYL719 completely blocks chondrogenesis in cultured hMSCs. In addition, in the histological analyses of these BYL719-treated mice that do not develop HO, we cannot find any sign of cartilage formation on either the 9th, 16th, or 23rd days post-injury, which confirms that BYL719 is also able to block chondrogenesis in mice in vivo. This suggests that intervention with BYL719 over a temporal window after an HO flare might be sufficient to inhibit endochondral ossification. In addition, although surgical resection is not recommended in FOP patients and has a risk of recurrence in resected non-genetic HO, it could be envisaged as preventive therapy in subjects undergoing surgeries to remove ectopic bone.

Treatment options for HO and FOP are currently limited, primarily consisting of anti-inflammatories such as corticosteroids and NSAIDs. Promising pharmacological treatments have progressed to II/III

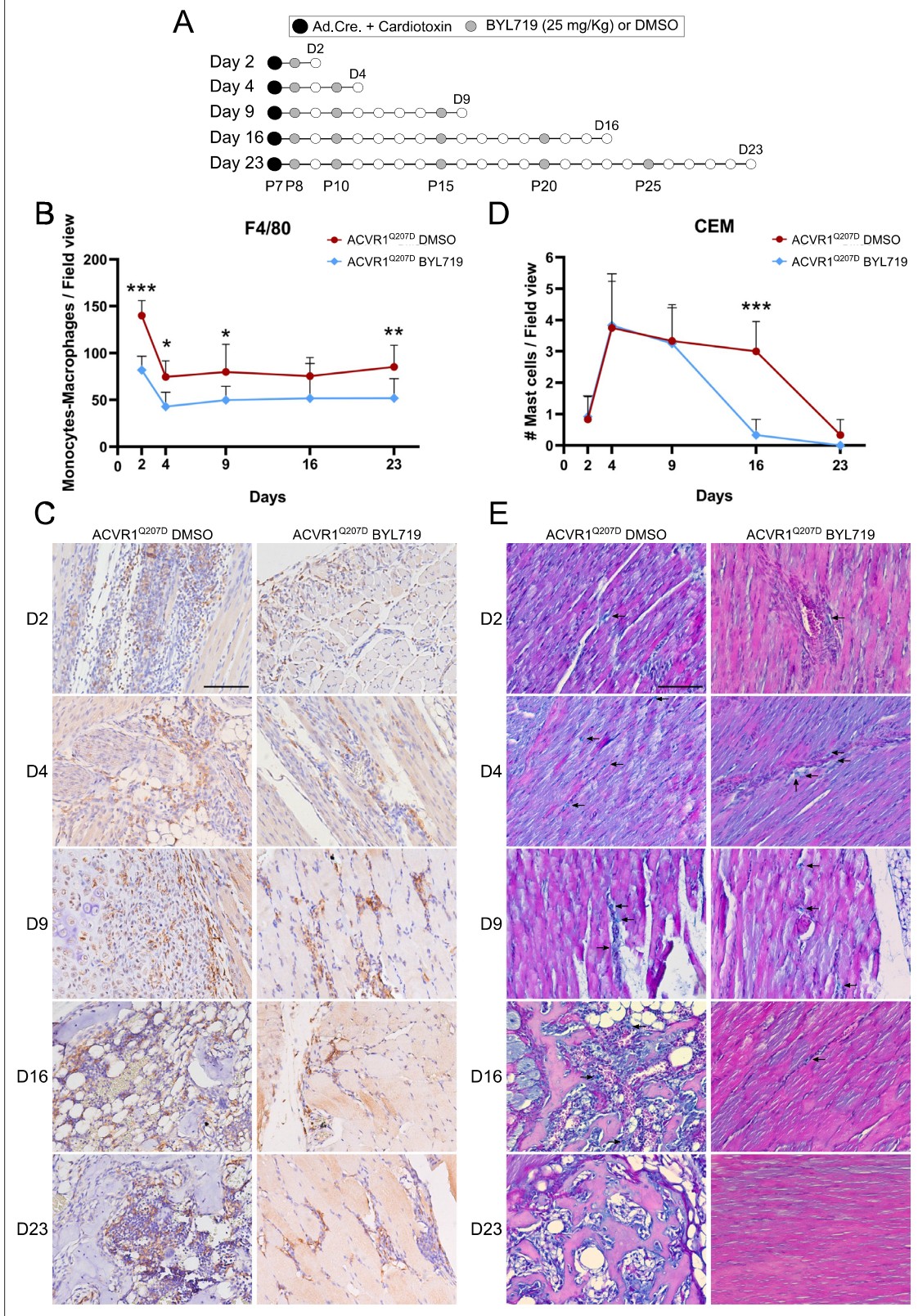

**Figure 7.** BYL719 reduces the number of monocytes and macrophages at injury sites. (**A**) Heterotopic ossification was induced at P7 through the injection of Adenovirus-Cre (Ad.Cre) and cardiotoxin in the mice hindlimb. Either DMSO (vehicle) or BYL719 (25 mg/kg) were injected following the scheme indicated with gray dots, starting at P8 but with a different duration P9 (2 days post-HO-induction), P11 (4 days post-HO-induction), P16 (9 days post-HO-induction), P23 (16 days post-HO-induction), and P30 (23 days post-HO-induction). (**B**) The quantification of F4/80-positive monocytes/

*Figure 7 continued on next page*

*Figure 7 continued*

macrophages per field view from immunohistochemistry staining, with ×20 amplification as shown in the representative images depicted in *Figure 8C*. Five images were randomly acquired per mice, from 2 mice per group, for a total of 10 quantified images per group. F4/80-positive cells were detected as cells with dark brown staining. Data are shown as mean ± SD. *p < 0.05, **p < 0.01, ***p < 0.001, two-way ANOVA with Sidak's multiple comparisons test, comparing both groups on each day (n = 10). (C) Representative images from the quantified immunohistochemistry staining for F4/80 to detect monocytes/macrophages. Representative images are shown for each final time point at day 2, 4, 9, 16, and 23 post-HO-induction. All images were acquired at ×20 (scale bar 100 μm). (D) Quantification of CEM-positive mast cells per field view from CEM staining, with ×20 amplification as shown in the representative images depicted in *Figure 8E*. Three images were obtained per mice, from 4 mice per group, for a total of 12 quantified images per group. Mast cells were detected as cells with bright blue staining. Data are shown as mean ± SD. ***p < 0.001, two-way ANOVA with Sidak's multiple comparisons test, comparing both groups in each day (n = 12). (E) C.E.M. staining was performed to detect mast cells, highlighted with black arrows. Representative images are shown for each final time point at day 2, 4, 9, 16, and 23 post-HO-induction. All the images were obtained at ×20 magnification, with a representative scale bar at 100 μm.

The online version of this article includes the following figure supplement(s) for figure 7:

**Figure supplement 1.** Time course of the BYL 719 inhibition of heterotopic ossification.

**Figure supplement 2.** Time course of the inhibition of chondrogenesis and osteogenesis by BYL719.

clinical trials. Palovarotene, a retinoic acid receptor-γ agonist, has been approved in Canada and USA for patients over 10 years old (NCT03312634). However, a previous clinical trial with palovarotene was paused for children due to concerns about premature growth plate closure, which is debated in FOP mice (*Chakkalakal et al., 2016*; *Rosen et al., 2018*). Antibodies targeting activin A (Garetosmab) showed effectiveness, but serious adverse effects led to a temporary hold on the trial (NCT03188666). The outcome of a clinical trial with Rapamycin has not been publicly disclosed, and a case report indicated limited benefits in classical FOP patients (UMIN000028429) (*Kaplan et al., 2018*). BYL719 (Alpelisib/Piqray) was recently licensed in combination with hormone therapy for *PIK3CA*-mutated breast cancer (*Cardoso et al., 2020*). BYL719 has been shown to be also clinically effective in adult patients with *PIK3CA*-related overgrowth syndrome (PROS) and children under 1 year of age (*Madsen and Semple, 2022*; *Morin et al., 2022*). In both scenarios, BYL719 had no major adverse effect when patients were treated for more than 1 year (in adults at a dose of 300 mg daily for oncological or PROS purposes and at a dose of 25 mg daily in infants with PROS) (*Morin et al., 2022*). Thus, implementing BYL719 for treatment of HO, at least during a restricted temporal window (i.e. surgery to remove ectopic bone in HO, or during flare-ups in FOP individuals), might be a valid therapeutic option for FOP patients.

## Materials and methods

### Murine bone marrow mesenchymal stem cells isolation

Murine BM-MSCs were isolated from 6- to 8-week-old C57BL/6J mice for the floxed *Pik3ca* allele (*Gámez et al., 2016*) as previously described (*Soleimani and Nadri, 2009*). We also isolated BM-MSCs from UBC-CRE-ERT2/ACVR1$^{R206H\ fl/wt}$ (a gift from Dr. Dan Perrien and IFOPA). This conditional ACVR1$^{R206H}$ construct was knocked-in to the endogenous *Acvr1* gene immediately following intron 4. After 4-OH tamoxifen addition, CRE activity excises murine *Acvr1* exons 5–10 and induce expression of the corresponding exons of human ACVR1$^{R206H}$ and an eGFP marker. In both cases, mice were euthanized and femurs were dissected and stored in DMEM with 100 U/ml penicillin/streptomycin (P/S). Soft tissues were cleaned and femur ends were cut under sterile conditions. Bone marrow was flushed with media using a 27-gauge needle. The resulting cell suspension was filtered through a 70-μm cell strainer and seeded in a 100-mm cell culture dish. Non-adherent cells were discarded after 3 hr. Media was slowly replaced every 12 hr for up to 72 hr. Then, media was replaced every 2 days until the culture reached 70% confluence. Then, cells were lifted by incubation with 0.25% trypsin/0.02% EDTA for 5 min at room temperature. Lifted cells were cultured and expanded. MSCs from UBC-CRE-ERT2/ACVR1$^{R206H\ fl/wt}$ mice were treated with 4-hydroxytamoxifen to induce Cre recombination.

### Production of hMSCs expressing ACVR1

For chondrogenic differentiation and RNA sequencing, bone marrow-derived human MSCs were transduced by lentiviral delivery encoding ACVR1$^{WT}$ or ACVR1$^{R206H}$ (*van Dinther et al., 2010*). For nanoBRET target engagement kinase assays, we were transduced with ACVR1$^{WT}$-nanoluc or ACVR1$^{R206H}$-nanoluc.

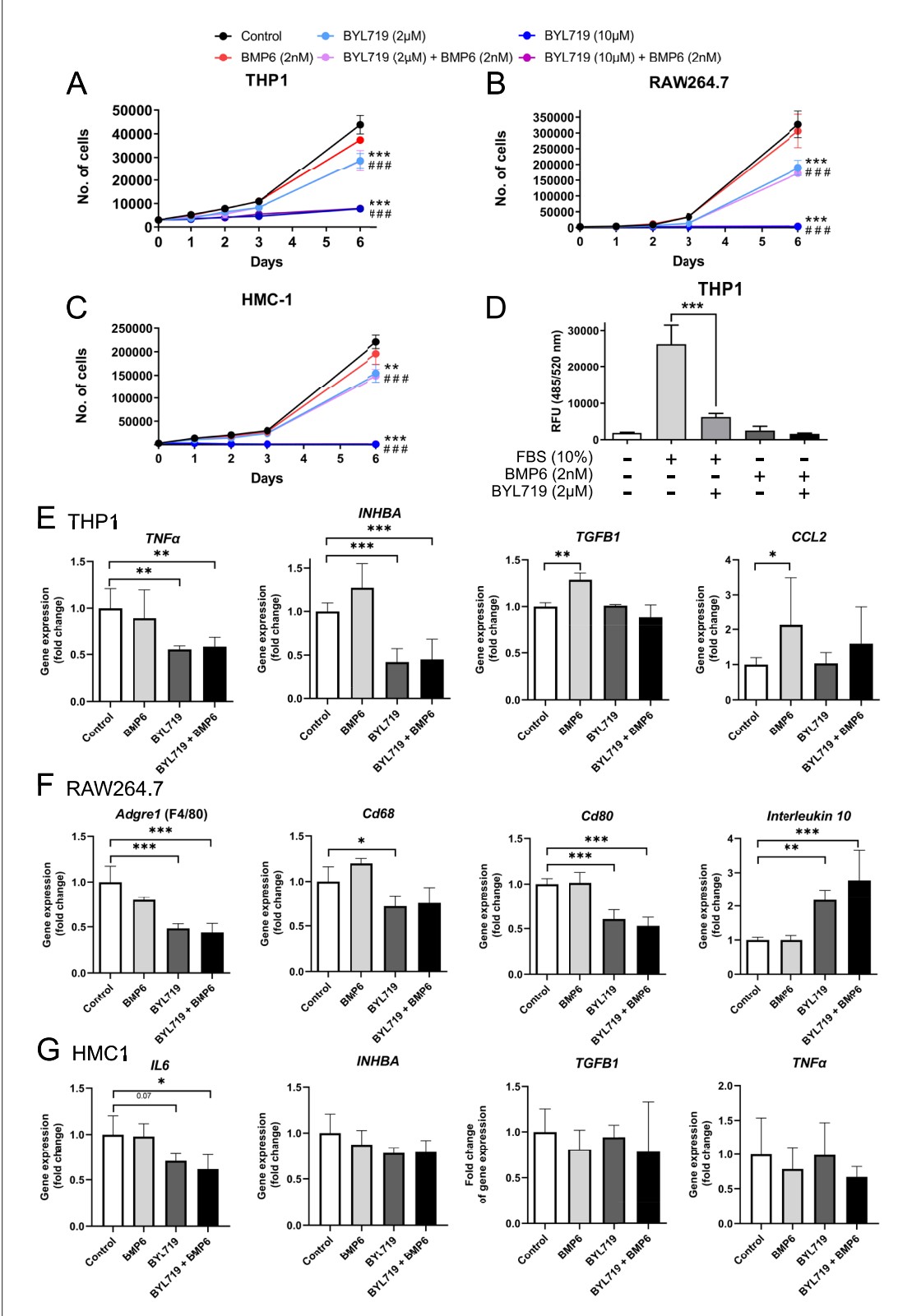

**Figure 8.** Proliferation assays of THP1 (**A**), RAW264.7 (**B**), and HMC-1 (**C**) cells. Cells were cultured for 6 days and in control conditions, with BMP6 (2 nM) and/or BYL719 (2 or 10 μM). Areas under the proliferation curves were compared by one-way ANOVA with Dunnett's multiple comparisons test against the control. (*) refers to significant differences between control and single treatments (BMP6 or BYL719). (#) refers to significant differences between control and combined treatments. Data are shown as mean ± SD (*n* = 4 per group). ** or p < 0.01, *** or ### p < 0.001. (**D**) Migration assay of THP1 cells

*Figure 8 continued on next page*

*Figure 8 continued*

with control conditions, or with FBS or BMP6 as chemotactic agents with or without BYL719 treatment. Data are shown as mean ± SD ($n$ = 4 per group). ***p < 0.001, two-way ANOVA with Tukey's multiple comparisons test. Gene expression assays of THP1 (**E**), RAW264.7 (**F**), and HMC-1 (**G**) cells. Cells were treated for 48 hr with BYL719 (2 μM) and/or BMP6 (2 nM). Data are shown as mean ± SD ($n$ = 4 per group). *p < 0.05, **p < 0.01, ***p < 0.001, one-way ANOVA with Dunnett's multiple comparisons test, significance shown between control group and other groups.

The online version of this article includes the following figure supplement(s) for figure 8:

**Figure supplement 1.** Proliferation assays of murine MSCs (**A**) and C2C12 cells (**B**).

The nanoluc fusion inserts (Promega, NV2341/NV2381) were subcloned into a PLV-CMV-IRES vector. First, an eGFP was amplified with PCR containing PstI, SalI, and XbaI restriction sites and subcloned in the PLV using PstI and XbaI. Next, the ACVR1-nanoluc inserts were subcloned in the PLV by restriction of PstI and SalI, exchanging the eGFP. Subsequently, lentiviral particles were produced using HEK293t cells and cells were transduced.

## Cell culture

Murine BM-MSCs were cultured in DMEM supplemented with 10% FBS, 2 mM glutamine, 1 mM sodium pyruvate, and 100 U/ml P/S and incubated at 37°C with 5% $CO_2$. The human monocyte cell line THP1 (ATCC) was cultured in RPMI-1640 supplemented with 10% FBS, 2 mM glutamine, 1 mM sodium pyruvate, 100 U/ml P/S, 0.1 mM NEAA, 10 mM HEPES, and 50 μM 2-mercaptoethanol. Mouse macrophage cell line RAW264.7 (ATCC) was cultured in DMEM supplemented with 10% FBS, 2 mM glutamine, 1 mM sodium pyruvate, and 100 U/ml P/S. The human mast cell line HMC1 (Sigma) was cultured in IMDM supplemented with 10% FBS, 2 mM glutamine, 1 mM sodium pyruvate, and 100 U/ml P/S. The hMSCs were cultured in αMEM supplemented with 10% FBS, 0.1 mM ascorbic acid, 1 ng/ml human bFGF, and 100 U/ml P/S (Growth Medium; GM). Murine C2C12 cells (ATCC) were cultured in DMEM supplemented with 20% FBS, 2 mM glutamine, 1 mM sodium pyruvate, and 100 U/ml P/S and incubated at 37°C with 5% $CO_2$. For C2C12 differentiation, cells were cultured for 7 days in the same media with 5% horse serum. All cell lines were periodically tested and remain negative for mycoplasma contamination.

## Proliferation and chemotactic migration assays

For proliferation assays, cells were seeded in 12-well culture plates in correspondent culture medium with 2 nM BMP6 and/or 2 μM or 10 μM BYL719 (Chemietek). Photos of several fields of each well were taken at 24, 48, 72, and 144 hr using a Leica DM-IRB inverted microscope. Cell counting was performed through ImageJ software.

The chemotactic migration assay of THP1 was performed through the InnoCyte Monocyte Migration Assay (Merck Millipore). Cells were serum-starved for 3 hr prior to the experiment and, when needed, treated with 2 μM BYL719 for 30 min before the experiment started. Then, $10^5$ cells/100 μl were added to each upper chamber. Lower chambers were charged with 10% FBS-supplemented RPMI-1640 (positive control), FBS-depleted RPMI-1640 (negative control) or 2 nM BMP6 (R&D), respectively. Cells migrated for 2 hr and were then stained with calcein-AM and pipetted into a 96-well conical bottom plate (Sigma-Aldrich). A plate reader was used to measure fluorescence from the top of the plate at an excitation wavelength of 485 nm and an emission wavelength of 520 nm. Unless otherwise stated, BYL719 at 2 or 10 μM (Chemietek) and 2 nM BMP6 (R&D) was used for 6 days (proliferation) or 4 hr (migration) in HMC1, THP1, and RAW264.7 cells.

## Retroviral transduction and gene expression analysis

BM-MSCs were infected with mock virus (pMSCV) or with viruses expressing wild-type *Acvr1* (WT), *Acvr1*[R206H] (RH), and/or pMSCV-Cre. Plasmids with WT and RH *Acvr1* forms were kindly provided by Dr. Petra Seemann and were subcloned into pMSCV vector. *Acvr1* and *Pik3ca* expression levels were analyzed by qRT-PCR. Cells were treated with BYL719 (2 μM) and/or activin A (2 nM) for 48 hr in complete media without FBS. THP1, RAW264.7, and HMC-1 cell lineswere treated in the media indicated for each cell line with BYL719 (2 μM) and/or BMP6 (2 nM) for 48 hr.

Total RNA for RT-PCR was extracted from all cellular models using TRIsure reagent. At least 2 μg of purified RNA was reverse-transcribed using the High-Capacity cDNA Reverse Transcription Kit

(Applied Biosystems). Quantitative PCRs were carried out on ABI Prism 7900 HT Fast Real-Time PCR System with Taqman 5'-nuclease probe method and SensiFAST Probe Hi-ROX Mix. All transcripts were normalized using *Tbp* as an endogenous control.

## NanoBRET target engagement assays

COS-1 cells (ATCC) were cultured in DMEM (Gibco, 11965092) supplemented with 10% FBS and 100 U/ml P/S. These cells were transfected in 70% confluent 6-well TC-treated plates with 2 µg TGF-β receptor–Nanoluciferase constructs (constructs were provided by Promega and Promega R&D) using a 1 µg DNA:2 µl PEI (1 mg/ml) ratio. The transfected cells were reseeded in white 384-well TC-treated assay plates 1 day before the nanoBRET readout in DMEM supplemented with 1% FBS in a quantity of $15 \times 10^3$ cells in 40 µl per well. Two hours before measurement, the dedicated tracer (Promega) and the test compounds were incubated to the cells at 37°C. Immediately after intracellular TE Nano-Glo Substrate/Inhibitor (Promega, N2160) addition, the wells were measured by 450-80BP and 620-10BP filters using the ClarioSTAR (BMG Labtech). Subsequent nanoBRET ratios were measured by the formula: acceptor emission (620–10 nm)/donor emission (450–80 nm) × 1000 (milliBRET units, mBU).

| Construct | Cat. # | Tracer (conc.) | Cat. # | Compounds (conc.) |
|---|---|---|---|---|
| *ACVRL1-nluc* | NV2391 | K-11 (0.31 µM) | N2652 | BYL719 (1 and 10 µM) LDN193189 (0.5 µM) |
| *ACVR1-nluc* | NV2341 | K-11 (0.16 µM) | N2652 | BYL719 (1 and 10 µM) LDN193189 (0.5 µM) |
| *ACVR1 R206H-nluc* | NV2381 | K-11 (0.16 µM) | N2652 | BYL719 (1 and 10 µM) LDN193189 (0.5 µM) |
| *BMPR1A-nluc* | NV2471 | K-11 (0.63 µM) | N2652 | BYL719 (1 and 10 µM) LDN193189 (0.5 µM) |
| *ACVR1B-nluc* | NV1021 | K-5 (2 µM) | N2482 | BYL719 (1 and 10 µM) SB431542 (10 µM) |
| *TGFBR1-nluc* | Promega R&D | K-14 (0.5 µM) | Promega R&D | BYL719 (1 and 10 µM) SB431542 (10 µM) |
| *ACVR2A-nluc* | Promega R&D | PBI6948 (1 µM) | Promega R&D | BYL719 (1 and 10 µM) ML347 (10 µM) |
| *ACVR2B-nluc* | Promega R&D | PBI6948 (1 µM) | Promega R&D | BYL719 (1 and 10 µM) ML347 (10 µM) |
| *TGFBR2-nluc* | Promega R&D | K-11 (0.32 µM) | N2652 | BYL719 (1 and 10 µM) ML347 (10 µM) |
| *BMPR2-nluc* | Promega R&D | PBI7394 (0.5 µM) | Promega R&D | BYL719 (1 and 10 µM) ML347 (10 µM) |

## Chondrogenic progenitor differentiation

MSCs were cultured in αMEM supplemented with 10% FBS, 1 ng/ml human bFGF, and 100 U/ml P/S (GM). Chondrogenic differentiation medium (DM) consisted of DMEM/F12 supplemented with 1% Insulin–Transferrin–Selenium, 170 µM ascorbic acid, 0.1 µM dexamethasone, 350 µM L-proline, 1 mM sodium pyruvate, 0.15% glucose, 1% FBS, and 100 U/ml P/S. During differentiation, the medium was refreshed twice a week for the duration of the experiment. For the Alcian Blue experiments, micro-mass cultures were made by carefully seeding $3 \times 10^5$ cells per 10 µl droplets in a 24-well plate and incubated for 2 hr at 37°C prior to GM addition. Chondrogenic progenitor differentiation was started 1 day after micromass seeding by the addition of DM. Dependent on the condition, we supplemented the DM with TGF-β1 (10 ng/ml) or activin A (100 ng/ml) and treated with BYL719 (10 µM) or

DMSO vehicle. For the alkaline phosphatase (ALP) staining, the hMSC-ACVR1 lines were seeded at $2 \times 10^4$ cells per well in 48-well plates and differentiated for 7 and 11 days.

## ALP staining, ALP activity assay, and Alcian Blue staining

For the ALP staining, the cells were washed with PBS, fixated using 3.7% formalin for 5 min at RT, washed twice with PBS and stained in ALP solution containing 2 mg Naphthol AS-MX, 6 mg Fast Blue, 5 ml 0.2 M Tris (pH 8.9), 100 µl $MgSO_4$, and $dH_2O$ up to 10 ml. Images were acquired using the Leica DMi8 with a ×10 magnification. To measure ALP activity, ALP lysates were obtained by washing twice with PBS, freezing at –80°C for 1 hr, and lysing on ice for 1 hr using 100 µl ALP buffer (100 µM $MgCl_2$, 10 µM $ZnCl_2$, 10 mM glycine (pH 10.5)) plus 0.1% Triton X-100 per well. ALP activity was quantified by adding 20 µl lysate and 80 µl 6 mM PNPP in ALP buffer to a clear 96-well plate incubated at RT until the samples turned yellow. Next, the samples were measured at an absorbance of 405 nm.

The micromass pellets were stained with Alcian Blue as described before (*Sánchez-Duffhues et al., 2019*). Images were acquired using the Leica DMi8 with ×5 magnification. Densitometry analysis of the whole well (from total plate scan) was performed using ImageJ (v1.53t).

## Bulk RNA sequencing and analyses

The hMSCs-ACVR1$^{WT}$ and hMSCs-ACVR1$^{R206H}$ were seeded at 100% confluency and serum-starved overnight with DMEM without supplements. Afterwards, the cells were pre-treated with or without 1 µM of BYL719 in starvation medium for 30 min and stimulated with or without 50 ng/ml activin A or 50 ng/ml BMP6 for 1 hr. RNA was isolated using the ReliaPrep RNA Miniprep Systems (Promega) and sequenced using the Illumina NovaSeq 6000 (Illumina) platform. The reads were mapped using the Hisat2 (v2.0.5) package and the total and normalized counts were measured using featureCounts (v1.5.0-p3). Differential expression analysis was performed using the DESeq2 R package (v1.20.0), in which the p-values were adjusted following the Benjamini and Hochberg approach. Genes with an adjusted p-value <0.05 were considered differentially expressed. Subsequent Gene Ontology analysis was performed using the clusterProfiler R package, and the GO terms were considered significant if the p-values <0,05. GSEA was performed using the GSEA analysis tool (gsea-msigdb.org). Heatmaps were generated using heatmap2 function within the R gplots package in Galaxy (usegalaxy.eu). The datasets are publicly available (see GSE237512).

## Western blot

Cells were lysed using RIPA lysis buffer and quantified using the Pierce BCA Protein assay (Thermo) following standard protocols. The protein samples were loaded using β-ME containing sample buffer and run using SDS–PAGE and transferred to PVDF membranes. Primary antibodies against SMAD1 (Cell Signaling, 6944), p-SMAD1/5 (Cell Signaling, 9516), and vinculin (Sigma, V9131) were used at 1:1000 dilution in 5% BSA in TBST. Binding was detected with HRP-conjugated secondary antibodies and visualized by Brightfield ECL (Thomas Scientific) on the ChemiDoc (Bio-Rad).

## In vitro recombinant kinase activity assay

Fifty ng of recombinant ACVR1$^{R206H}$ (ab167922, Abcam) was diluted in 30 µl of 25 mM Tris/HCl (pH 7.5), 10 mM $MgCl_2$, 10 mM $MnCl_2$, and 2 mM DTT. The reaction was started by the addition of 5 µM ATP-Mg, 0.05 µCi/µl ATP [γ-$^{32}$P], and 0.5 mg/ml of dephosphorylated casein (Sigma) and performed for 30 min at 30°C. The reaction was stopped, and samples were processed by SDS–PAGE. After drying the gel, phosphorylation was determined by autoradiography.

## HO mouse model

To study HO in vivo, we used the Cre-inducible constitutively active ACVR1$^{Q207D}$ (CAG-Z-EGFP-caALK2) mouse model as previously described (*Fukuda et al., 2006*; *Shimono et al., 2011*; *Yu et al., 2008*). Mice ACVR1Q207D$^{fl/fl}$ Pik3ca$^{fl/fl}$ were obtained by crossing the detailed homozygous ACVR1Q207D$^{fl/fl}$ with the Pik3ca$^{fl/fl}$ homozygous mutant mice carrying loxP sites flanking exons 18 and 19 of the Pik3ca alleles (*Gámez et al., 2016*). Heterozygous mice were crossed until ACVR1Q207D$^{fl/fl}$ Pik3ca$^{fl/fl}$ was obtained.

To induce HO in P7 ACVR1$^{Q207D}$ mice, $1 \times 10^8$ pfu of Adenovirus-Cre (Ad-CMV-Cre, Viral Vector Production Unit, UAB) and 0.3 µg of cardiotoxin in 10 µl 0.9% NaCl volume were injected into the

left hindlimb. Control groups had the same procedure without Ad-Cre in the injection. On P8, mice started receiving either placebo (intraperitoneal administration (i.p.) of DMSO) or BYL719 (i.p. of 25 mg/kg), both diluted in 0.5% carboxymethylcellulose sodium. Mice were housed under controlled conditions (12-hr light/12-hr dark cycle, 21°C, 55% humidity) and fed ad libitum with water and a 14% protein diet (Teklad2014, Envigo). Mice were regularly weighed over the whole period. HO induction was performed blinded to mouse and group identity. All procedures were approved by the Ethics Committee for Animal Experimentation of the Generalitat de Catalunya (#336/19, #195/22, and #11110).

## Micro-computed tomography analysis

The caudal half of mice was collected and fixed in 4% paraformaldehyde (PFA) for 48 hr at 4°C. Samples were conserved in PBS, and high-resolution images were acquired using a computerized microtomography imaging system (Skyscan 1076, Bruker microCT), in accordance with the recommendations of the American Society of Bone and Mineral Research (ASBMR). Samples were scanned in air at 50 kV and 200 µA with an exposure time of 800 ms, using a 1-mm aluminum filter and an isotropic voxel size of 9 µm. Two-dimensional images were acquired every 1° for 180° rotation and subsequently reconstructed, analyzed for bone parameters, and visualized by NRecon v1.6, CT-Analyser v1.13, and CTVox v3.3 programs (Bruker), respectively. For HO, manual VOIs comprising HOs were employed, and a binary threshold was established at 25-255.

## Histology and immunofluorescence

Whole legs were fixed in 4% PFA for 48 hr at 4°C, decalcified in 16% EDTA pH 7.4 for 6 weeks, and embedded in paraffin. Five µm sections were cut and stained with Fast Green/Safranin O, or Masson's Trichrome. Images were obtained with brightfield Eclipse E800 (Nikon). To detect monocyte/macrophage cells, paraffined slides were immunostained with Anti-F4/80 antibody [SP115] (ab111101). Briefly, after deparaffinization and rehydration, an antigen retrieval step with citrate buffer at pH 6 in a decloaking chamber was applied to the slides. Slides were cooled down and rinsed with 1× TBS and endogenous peroxidases were blocked with 70% methanol, 28% distilled water, and 2% hydrogen peroxide for 5 min at room temperature. After a blocking step, F4/80 primary antibody incubation at concentration 1:100 overnight at 4°C was applied. Envision Dual Link conjugated with HRP was applied for 40 min at room temperature, and detection was performed with diaminobenzidine for 3 min. Nuclear staining with Lilly's hematoxylin was applied to the slides for 30 s. Images of stained slides were obtained with brightfield Eclipse E800 (Nikon). Images were randomly acquired with ×20 magnification from the region of interest (injected muscle, with or without visible HO). Five images were obtained per mice, from 2 mice per group, for a total of 10 quantified images per group. F4/80-positive cells were detected as cells with dark brown staining. To detect mast cells, paraffined slides were stained with C.E.M. staining using the Eosinophil-Mast cell staining kit (ab150665, Abcam). The recommended staining procedure was applied to the deparaffinized sections. Images of stained slides were obtained with brightfield Eclipse E800 (Nikon). Images were randomly acquired with ×20 magnification from the region of interest (injected muscle, with or without visible HO). Three images were obtained per mice, from 4 mice per group, for a total of 12 quantified images per group. Mast cells were detected as cells with bright blue staining.

For immunofluorescence assays, 5 µm sections of each time point and condition were deparaffinased, processed with antigen retrieval and permeabilization steps, and stained with wheat germ agglutinin (1:250 dilution, Thermo Fisher Scientific), followed by incubation using PDGFRA antibody (1:250 dilution Cell Signalling #3174), Alexa555 secondary antibody, and DAPI staining. Images were randomly acquired with ×20 magnification from the region of interest with a Carl Zeiss Axio Imager M2 Apotome microscope. Five images were obtained per mice, from 4 mice per group, for a total of 20 quantified images per group.

## Statistical analysis

Unless stated otherwise in each figure legend, the results were expressed as mean ± SD. The median was shown in HO datasets where non-parametric tests were applied. Each figure legend has detailed information explaining the statistical test used to compare between groups, the compared groups, the explanation of the symbols used to show significance, usually *p < 0.05, **p < 0.01, ***p < 0.001,

and the sample size of the experiment. HO, evaluation, and quantification were performed blinded to mouse and group identity. Microscopy and histology representative images were selected from the total quantified images. Microscopy and histology quantified fields were randomly acquired from the plate or the region of interest. Statistical tests were performed on GraphPad Prism 9.5. Any material is available upon request.

## Acknowledgements

We thank E Adanero, E Castaño for technical assistance. We acknowledge the support from Promega R&D. Alexandre Deber is a recipient of an FPI fellowship from the Ministry of Science and Innovation (MCIN/AEI/10.13039/501100011033). Carolina Pimenta-Lopes is the recipient of a FPU fellowship from the Spanish Ministry of Education. This research was supported by grants PID2022-141212OA-I00, PDC2021-121776-I00, and PID2020-117278GB-I00 from MCIN/AEI/10.13039/501100011033, co-funded by FEDER 'Una manera de hacer Europa' and 'NextGenerationEU'/PRTR; a grant 202038-30 from La Marató de TV3 and grants from IFOPA (ACT for FOP) and FOP Italia. MW and MJG are sponsored by the Netherlands Cardiovascular Research Initiative (the Dutch Heart Foundation, Dutch Federation of University Medical Centers, the Netherlands Organization for Health Research and Development, and the Royal Netherlands Academy of Sciences), PHAEDRA-IMPACT (DCVA), and DOLPHIN-GENESIS (CVON). GSD is also sponsored by the Spanish Ministry of Science through the Ramón y Cajal grant RYC2021-030866-I and the BHF-DZHK-DHF, 2022/23 award PROMETHEUS.

## Additional information

### Funding

| Funder | Grant reference number | Author |
| --- | --- | --- |
| Spanish National Plan for Scientific and Technical Research and Innovation | PID2023-148874NB-I00 | Francesc Ventura |
| Spanish National Plan for Scientific and Technical Research and Innovation | PDC2021-121776-I00 | Francesc Ventura |
| La Marató de TV3 | 202038-30 | Francesc Ventura |
| Ministry of Science and Innovation | MCIN/AEI/10.13039/501100011033 | Alexandre Deber |
| Spanish Ministry of Education | PID2022-141212OA-I00 | Gonzalo Sánchez-Duffhues |
| Spanish Ministry of Education | PID2020-117278GB-I00 | Francesc Ventura |
| Spanish Ministry of Science through the Ramón y Cajal | BHF-DZHK-DHF, 2022/23 | Gonzalo Sánchez-Duffhues |
| Spanish Ministry of Science through the Ramón y Cajal | RYC2021-030866-I | Gonzalo Sánchez-Duffhues |

The funders had no role in study design, data collection, and interpretation, or the decision to submit the work for publication.

### Author contributions

José Antonio Valer, Conceptualization, Data curation, Formal analysis, Methodology, Writing – original draft, Writing – review and editing; Alexandre Deber, Conceptualization, Data curation, Formal analysis, Investigation, Writing – original draft, Writing – review and editing; Marius Wits, Investigation, Writing – original draft; Carolina Pimenta-Lope, Formal analysis, Investigation, Methodology; Marie-José Goumans, Jose Luis Rosa, Supervision; Gonzalo Sánchez-Duffhues, Data curation, Supervision, Investigation, Methodology, Writing – original draft, Writing – review and editing; Francesc

Ventura, Conceptualization, Data curation, Formal analysis, Supervision, Funding acquisition, Investigation, Methodology, Writing – original draft, Writing – review and editing

### Author ORCIDs
José Antonio Valer ![ORCID] https://orcid.org/0000-0003-1184-9491
Jose Luis Rosa ![ORCID] https://orcid.org/0000-0002-6161-5688
Gonzalo Sánchez-Duffhues ![ORCID] https://orcid.org/0000-0002-3205-0710
Francesc Ventura ![ORCID] https://orcid.org/0000-0001-9673-9405

### Ethics
All procedures for animal handling and experimentation were approved by the Ethics Committee for Animal Experimentation of the Generalitat de Catalunya (#336/19, #195/22, and #11110).

Reviewer #1 (Public review): https://doi.org/10.7554/eLife.91779.4.sa1
Reviewer #2 (Public review): https://doi.org/10.7554/eLife.91779.4.sa2
Author response https://doi.org/10.7554/eLife.91779.4.sa3

---

## Additional files

### Supplementary files
MDAR checklist

### Data availability
Sequencing data has been deposited in GEO under accession code GSE237512.

The following dataset was generated:

| Author(s) | Year | Dataset title | Dataset URL | Database and Identifier |
|---|---|---|---|---|
| Wits M, Goumans M, Ventura F, Sánchez-Duffhues G | 2024 | PI3Kα inhibition blocks osteochondroprogenitor specification and the hyper-inflammatory response to prevent heterotopic ossification | https://www.ncbi.nlm.nih.gov/geo/query/acc.cgi?acc=GSE237512 | NCBI Gene Expression Omnibus, GSE237512 |

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
