## [Editor Report · eLife Assessment]

This study, which includes additional experiments in response to the reviewer comments, presents **valuable** findings illustrating the role of PI3Kα in heterotopic ossification in FOP model mice. The methods, data, and analyses are **solid** and generally support the claims although as noted by one of the reviewers, there is no data demonstrating the effect of BYL79 on cell growth, and it remains unclear whether BYL79 also inhibits the Smad2/3 pathway. While this study provides new insights into the role of the PI3Kα pathway as a therapeutic target for FOP, questions about the mechanism of BYL79 still exist.

---

## [Referee Report · Reviewer #1 (Public review)]

Summary:

In the present study, the authors examined the possibility of using phosphatidyl-inositol kinase 3-kinase alpha (PI3Ka) inhibitors for heterotopic ossification in fibrodysplasia ossificans progressiva. Administration of BYL719, a chemical inhibitor of PI3Ka, prevented heterotopic ossification in a mouse model of FOP that expressed a mutated ACVR1 receptor. Genetic ablation of PI3Ka also suppressed heterotopic ossification in mice. BYL719 blocked osteo/chondroprogenitor specification and reduced inflammatory responses by reducing the number of fibro-adipogenic progenitors (FAPs) and promoting muscle fibre regeneration in vivo. The authors claimed that inhibition of PI3Ka is a safe and effective therapeutic strategy for heterotopic ossification.

Strengths:

Taking together previous reports on the specificity of BY718 in PI3K, it was suggested that BYL719 inhibits heterotopic ossification by reducing FAPs and promoting muscle regeneration through the PI3K pathway in vivo.

Weaknesses:

In the original manuscript, there was the possibility that BYL719 inhibited heterotopic ossification through non-specific and toxic effects rather than the PI3k pathway.

However, the authors added new data and explanations in the revision to solve the possibility. The findings of the authors would be useful and would provide an additional direction to develop a therapeutic strategy for heterotopic ossification.

---

## [Referee Report · Reviewer #2 (Public review)]

Summary:

Authors in this study previously reported that BYL719, an inhibitor of PI3Kα, suppressed heterotopic ossification in mice model of a human genetic disease, fibrodysplasia ossificans progressive, which is caused by the activation of mutant ACVR1/R206H by Activin A. The aim of this study is to identify the mechanism of BYL719 for the inhibition of heterotopic ossification. They found that BYL719 suppressed heterotopic ossification in two ways: one is to inhibit the specification of precursor cells for chondrogenic and osteogenic differentiation and the other is to suppress the activation of inflammatory cells.

Strengths:

This study is based on authors' previous reports and the experimental procedures including the animal model are established. In addition, to confirm the role of PI3Kα, authors used the conditional knock-out mice of the subunit of PI3Kα. They clearly demonstrated the evidence indicating that the targets of PI3Kα is not members of TGFBR by a newly established experimental method.

Weaknesses:

Overall, the presented data were closely related to those previously published by authors' group or others and there were very few new findings. The molecular mechanisms through which BYL719 inhibits HO remain unclear, even in the revised manuscript.

Heterotopic ossification in mice model was not stable and inappropriate for the scientific evaluation.

The method for chondrogenic differentiation was not appropriate, and the scientific evidence of successful differentiation was lacking.

The design of gene expression profile comparison was not appropriate and failed to obtain the data for the main aim of this study.

The experiments of inflammatory cells were performed cell lines without ACVR1/R206H mutation, and therefore the obtained data were not precisely related to the inflammation in FOP.

Comments on revisions:

In the R2 version, the authors performed additional experiments using mice with inducible human R206H ACVR1A. BM-MSCs isolated from these mice were used to investigate the effect of Activin-A. The results again suggested that BYL79 inhibited the chondrogenic differentiation of BM-MSCs. However, there are still no data demonstrating the effect of BYL79 on cell growth in these in vitro experiments. In Figures 7A-D, 10 μM BYL79 strongly inhibited the proliferation of inflammatory cells, suggesting that growth inhibition may have contributed to the results shown in Figure 5.

The main point of discussion concerns the significance of the comparisons made. The fundamental disagreement arises from the role of Activin-A in R206H cells and its effect on chondrogenic differentiation. The authors' rebuttal regarding my comments on the RNA-seq analyses should be reconsidered. The core issue lies in the interpretation of Activin-A's role in R206H cells and the distinction between chondrogenic differentiation and ossification.

A key feature of R206H mutant cells is that they respond to Activin-A by activating Smad1/5 signaling-comparable in quality to the signaling induced by BMP6 in WT cells. Another important point, as also acknowledged by the authors, is that Activin-A can transduce Smad2/3 signaling via its canonical receptor, ACVR1B. These dual signaling pathways synergistically contribute to chondrogenic differentiation in precursor cells such as FAPs. Several reports have demonstrated that the combined activation of TGF-β and BMP signaling promotes chondrogenesis more strongly than either pathway alone.

Since the PI3Kα inhibition effect on HO is already known, a critical question in this study is whether BYL79 also inhibits the Smad2/3 pathway. A straightforward experiment would be to compare WT cells treated with Activin-A alone versus Activin-A plus BYL79, and to perform GO term enrichment analyses related specifically to chondrogenic differentiation, not ossification. Additionally, comparing R206H cells treated with Activin-A/BYL79 and WT cells treated with BMP6/BYL79 could help identify gene sets inhibited by BYL79 via Smad2/3 signaling. If these comparisons reveal no specific effect on genes related to chondrogenesis, the effect of BYL79 may be limited to suppression of BMP-mediated osteogenesis. Unfortunately, the authors appear to show little interest in addressing this issue.

Regarding Figure 7, the authors' rebuttal should also be reconsidered. Since the R2 version employed FOP model mice, it would have been possible to evaluate the effects of BYL79 on inflammatory cells harboring the R206H mutation. This could have enabled a more precise assessment of BYL79's influence on inflammatory signaling. While the authors repeatedly claim that BYL79's effect is not specific to any particular ligand or the presence of the FOP mutation, the role of TGF-β signaling in the development of endochondral heterotopic ossification is well recognized. Therefore, the mechanism of BYL79 should be clarified before considering its therapeutic application

---

## [Author Response]

The following is the authors’ response to the previous reviews

Our revised manuscript thoroughly addresses all comments and suggestions raised by the reviewers, as detailed in our point-by-point response. To strengthen our findings, we have conducted additional in vivo experiments to evaluate the presence of fibro-adipogenic progenitors (FAPs) at different time points during HO formation in control and BYL719-treated mice. Our results indicate that BYL719 reduces the accumulation of FAPs and promotes muscle fiber regeneration in vivo. We have also expanded our discussion on BYL719’s effects on mTOR signaling, further clarifying key points raised by Reviewer #1, and have addressed all minor comments.

Additionally, in response to Reviewer #2, we have employed an orthogonal and complementary approach using a new model. We conducted chondrogenic differentiation experiments with murine MSCs expressing either ACVR1wt or ACVR1^R206H^. qPCR analysis of chondrogenic gene markers (*Sox9, Acan, Col2a1*) demonstrates that Activin A enhances their expression in ACVR1^R206H^ cells, whereas BYL719 strongly suppresses their expression, regardless of ACVR1 mutational status. These new data further confirm that BYL719 effectively inhibits genes involved in ossification and osteoblast differentiation, independent of the ACVR1 mutation. We have also expanded our discussion to further clarify points raised by Reviewer #2 and have addressed all remaining minor comments.

Below, we provide a detailed point-by-point response to the reviewers’ comments:

**Rreviewer #1:**
Point 1: In this revised manuscript, the authors clearly showed that BYL719 suppressed the proliferation and differentiation of murine myoblasts, C2C12 cells, in addition to human MSCs in vitro. Furthermore, BYL719 decreased migratory activity in vitro in monocytes and macrophages without suppressing proliferation. Overall, these data suggested that BYL719 is not a specific chemical compound for cell types or signaling pathways as mentioned in the manuscript by the authors themselves. Therefore, it was still unclear how to explain the molecular mechanisms in inhibition of HO by the compound in a specific signaling pathway in a specific cell type, MSCs, contradicting many other possibilities. The authors should add logical explanations in the manuscript.

Regarding its selectivity, BYL719 is a potent and highly selective inhibitor of PI3Kα. It has been demonstrated in multiple studies and in several in vitro kinase assay panels (Furet et al. PMID: 23726034, Fritsch et al. PMID: 24608574). The IC50 or Kd values for BYL719 against PI3Kα were at least 50 times lower than for most of other kinases tested. Moreover, BYL719 is also highly selective for PI3Kα (IC50 = 4.6 nmol/L) compared to other class I PI3K (PI3Kβ (IC50 = 1,156 nmol/L), PI3Kδ (IC50 = 290 nmol/L), PI3Kγ (IC50 = 250 nmol/L)) (Fritsch et al). Consistent with these data, we show that, at the concentrations tested, BYL719 does not have a direct effect on any kinase receptor within the TGF-b superfamily, including ACVR1 or ACVR1^R206H^.

Rather than blocking ACVR1 kinase activity, in our manuscript we provide evidence that BYL719 has the potential to inhibit osteochondroprogenitor specification and prevent an exacerbated inflammatory response in vivo (Valer et al., 2019a PMID: 31373426, and this manuscript) through different mechanisms, such as (i) increasing SMAD1/5 degradation, (ii) reducing transcriptional responsiveness to BMPs and Activin, (iii) blocking non-canonical ACVR1 responses such as the activation of AKT/mTOR. All these defined molecular mechanisms contribute to suppress HO in vitro and in vivo, as we report and explain throughout the manuscript. Selective PI3Kα inhibition is at the core of the different molecular pathways described. As such, PI3Kα blockade inhibits the phosphorylation of GSK3 and compromises SMAD1 protein stability, thereby altering canonical responsiveness and osteochondroprogenitor specification (Gamez et al PMID: 26896753; Valer et al PMID: 31373426). Moreover, PI3Kα blockade downregulates Akt/mTOR signalling, which is critical for FOP and non‐genetic (trauma induced) HO in preclinical models (Hino et al, 2017 PMID: 28758906; Hino et al. PMID: 30392977). Finally, PI3Kα inhibition hampers a number of proinflammatory pathways, thereby limiting the expression of pro-inflammatory cytokines, reducing the proliferation of monocytes, macrophages and mast cells, and partially blocking the migration of monocytes. As we suggest in the discussion of the manuscript, this effect likely causes a poor recruitment of monocytes and macrophages at injury sites and throughout the in vivo ossification process.

Noteworthy, in our manuscript we do not refer to a “specific chemical compound for cell types”. Rather, in the Discussion we write “the administration of BYL719 prevented an exacerbated inflammatory response in vivo, possibly due to specific effects observed on immune cell populations.” This sentence did not intend to imply that BYL719 only affects these specific cell types, but aimed to emphasize the effects observed on those cell populations, even though systemic BYL719 may affect all populations. We rephrased it to “the administration of BYL719 prevented an exacerbated inflammatory response in vivo, possibly due to the effects observed on immune cell populations.” to provide a clearer message as suggested by the reviewer. We thank the reviewer for these questions and hope that these explanations and changes in the text improve the clarity of the message.

Mesenchymal stem/stromal cells (MSCs) are osteochondroprogenitor cells that can follow distinct differentiation paths. In this study, we use these cells as an in vitro model for the study of osteochondrogenitor specification. MSCs, and induced MSCs (iMSCs), have been widely used as in vitro cellular models of osteochondroprogenitor specification for the analysis of markers, signaling, modulation, and differentiation potential or capacity. Their use as models for this purpose has been extensively studied in wild type MSCs, and in the presence of FOP mutations (Boeuf and Richter PMID: 20959030; Schwartzl et al. PMID: 37923731).

Point 2: Related to comment #1, the effects of BYL719 on the proliferation and differentiation of fibro-adipogenic cells in skeletal muscle, which are potential progenitor cells of HO, should be important to support the claim of the authors.

We have performed additional in vivo experiments to assess the presence of fibro-adipogenic precursors (FAPs) at different time-points during HO formation in control and BYL719-treated in the mouse model of heterotopic ossification. We analyzed the number of fibro-adipogenic progenitor (FAPs) during the progression of the HO. These data are shown in the new Figure3-Figure Supplement 1. We demonstrate that BYL719 reduces the number of PDGFRA+ cells (FAPs, red) throughout the ossification process in vivo. Moreover, now we also show an enlargement of the diameter of myofibers (labelled with wheat germ agglutinin, green) when animals were treated with BYL719, indicating improved muscle regeneration and further validating the data reported as supplementary figures that were added in the first revision of this manuscript.

Point 3: BYL719 inhibited signaling through not only ACVR1-R206H and ACVR1-Q207D but also wild type ACVR1 and suppressed the chondrogenic differentiation of parental MSCs regardless of the expression of wild type or mutant ACVR1. Again, these findings suggest that BYL719 inhibits HO through a multiple and nonspecific pathway in multiple types of cells in vivo. The authors are encouraged to explain logically the use of bone marrow-derived MSCs to examine the effects of BYL719.

As detailed in main point 1, we consider that the main target, molecular mechanisms and inhibited pathways by BYL719 are specific and well characterised in other research articles and further defined in this manuscript, including the generation of PI3Ka deficient mice in an FOP background, that undoubtedly demonstrates an essential role for PI3Ka in ACVR1-driven heterotopic ossification in vivo. Altogether, we are confident that BYL719 inhibits HO through multiple and specific pathways that arise from the PI3Kα inhibition. As a systemically administrated drug, BYL719 affects the multiple types of cells in vivo that express PI3Kα. It is well known that PI3Kα is exquisitely required for chondrogenesis and osteogenesis (Zuscik et al. PMID; Gamez et al PMID: 26896753 1824619). Accordingly, throughout the manuscript we refrain from suggesting a specific effect on ACVR1-R206H cells but instead an inhibitory effect on cell number and differentiation regardless on the ACVR1 form expressed.

Similarly, as detailed in main point 1, MSCs and hiPSCs have been extensible used as in vitro cellular models of osteochondroprogenitor specification for the analysis of markers, signaling, modulation, and differentiation potential or capacity (Barruet et al., PMID: 28716551; Kan et al., PMID: 39308190).

Point 4: BYL719 clearly inhibits an mTOR pathway. Is there a possibility that BYL719 suppresses HO by inhibiting mTOR rather than PI3K? The authors are encouraged to show the unique role of PI3K in BYL719-suppressed HO formation.

As clarified above, BYL719 is a potent and selective inhibitor of PI3Kα, with minimal off-target inhibition against other kinases, as it has been demonstrated in multiple studies and in several *in vitro* kinase assay panels. In the same study, while IC50 of BYL719 against PI3Kα was (IC50 = 4.6 nmol/L), IC50 against mTOR was (IC50 = >9,100 nmol/L), indicating that it was not directly inhibited. mTOR is one of the well-known pathways that are activated downstream of PI3K. Therefore, there is no surprise that blocking PI3Kα will block mTOR signalling. This potential effect was already demonstrated in previous publications (Valer et al., 2019a PMID: 31373426) and discussed throughout the first revision. We consider that the additive effect of mTOR inhibition and other molecular mechanisms downstream of PI3Kα, including reduced SMAD1/5 protein levels, contribute to the in vivo HO inhibition by BYL719.

**Reviewer #2:**
Point 1: It is also important to note that, in most of the data, there is no significant difference between cells with wild-type ACVR1 and those with the R206H mutation. The authors demonstrated that ACVR1 is not a target of BYL719 based on NanoBRET assay data, suggesting that BYL719's effect is not specific to FOP cells, even though they used an FOP mouse model to show in vivo effects.

The main effect of R206H mutation is the gain of function in response to Activin A. For most of the responses to other ACVR1 ligands (e.g. BMP6/7), we observe a slightly increased response in the presence of the mutation (which is consistent with previous research, usually labelling RH as a “weak activating mutant” unless Activin A is added (Song et al., PMID: 20463014)). Therefore, as expected, most of the differences between WT and RH mutant cells can be observed mostly upon Activin A addition, as observed, for example, in Figure 3 of our manuscript.

We agree with the reviewer that, at the concentrations used, BYL719 does not specifically target FOP cells. However, we believe that it targets downstream pathways of PI3Kα inhibition that are essential for osteochondrogenic specification, regardless of mutation status. This therapeutic strategy aligns with other experimental drugs, including Palovarotene (validated for FOP) and Garetosmab and Saracatinib (in advanced clinical trials), which target Activin A function, ACVR1 activity, or osteochondrogenic differentiation irrespective of the mutant allele. Unlike these molecules, BYL719 has been chronically administered to patients (including children) without major side effects (Gallagher et al.; PMID: 38297009), further supporting its potential for safe long-term use.

The authors should consider that the effect of Activin A on R206H cells is not identical to that of BMP6 on WT cells. If the authors aim to identify the target of BYL719 in FOP cells, they should compare R206H cells treated with Activin A/BYL719 to WT cells treated with BMP6/BYL719.

We use Activin A and BMP6, both high-affinity ACVR1 ligands, to demonstrate, as observed in figure 6, that PI3Kα inhibition can inhibit the expression of genes within GO terms ossification and osteoblast differentiation. It is important to note, however, that Activin A canonical signaling receptor is ACVR1B. Since BYL719 blocks the induction of a heterotopic ossification gene expression signature common to Activin A and BMP6, in the context of the FOP mutation R206H, our results indicate that BYL719 inhibition affects a signaling pathway downstream of ACVR1, activated by either BMP6 (wild type receptor, relevant for non-genetic heterotopic ossifications) or Activin (R206H mutant receptor, relevant for FOP).

We consider that the comparison (RH ACTA BYL vs WT BMP6 BYL) would provide confounding results raised from intrinsic model differences in basal expression programs (WT vs RH), and differences in the quantitative level of signaling of the different ligands at these specific doses. First, if we only consider SMAD1/5 signaling, Activin A and BMP6 won’t have identical signaling, and differences will arise from the strength of that signaling. Secondly, in the suggested comparison we would find, mostly, all the differential gene expression promoted by Activin A canonical signaling through type I receptors ACVR1B/ALK4 in complex with ACVR2A or ACVR2B, promoting SMAD2/3 activation (in addition to the altered signaling that ACVR1-R206H could promote). Examples of differential response in pSMAD1/5 in ACVR1-WT or RH with BMP ligands and R206H with Activin A ligand, and examples of pSMAD2/3 canonical signaling in R206H cells have been described in Ramachandran et al, PMID: 34003511; Hatsell et al., PMID: 26333933.

Point 2: The interpretation of the data in the new Figure 5 is inappropriate. Based on the expression levels of SOX9, COL2A1, and ACAN, it is unclear whether the effect of BYL719 is due to the inhibition of differentiation or proliferation. The addition of Activin A showed no difference between ACVR1/WT and ACVR1/R206H cells, suggesting that these cells did not accurately replicate the FOP condition.

To gain consistency in our manuscript, we decided to use an orthogonal and complementary approach in a completely new model. We performed new experiments of chondrogenic differentiation using murine MSCs from UBC-Cre-ERT2/ACVR1^R206H^ knock-in mice. These cells, when treated with 4OH-tamoxifen, express the intracellular exons of human ACVR1^R206H^ in the murine *Acvr1* locus. Therefore, we can compare differentiation of wild type and R206H MSCs isolated form the same mice. We initiated the chondrogenic differentiation assay from confluent cells to minimize changes in cell proliferation throughout the process. These new results are shown in the new Figure 5F. Mutant (RH) cells display an enhanced chondrogenic response to activin A compared to wild type cells. The treatment with BYL719 decreased the expression of chondrogenic markers irrespective of the mutational status of ACVR1 in the cells, further supporting our previous results in this manuscript and published article (Valer et al., 2019a PMID: 31373426).

Point 3: The additional investigation of RNA-seq data provided useful information but was insufficient to fully address the purpose of this study. The authors should identify downregulated genes by comparing WT cells treated with Activin A/BYL719 and Activin A alone and then compare these identified genes with those shown in Figure 5E. Additionally, they should compare R206H cells treated with Activin A/BYL719 to WT cells treated with BMP6/BYL719. These comparisons will clarify whether there are FOP-specific BYL719-regulated genes.

We thank the reviewer for considering that RNAseq data provides useful information. As already discussed in our answer above, our results indicate that regardless of the ligand (Activin A or BMP6) and regardless of the ACVR1 mutation (WT, relevant for non-genetic heterotopic ossifications or RH, relevant for FOP), BYL719 can inhibit the expression of the genes relevant to endochondral ossification. In our opinion, this is a very relevant conclusion of this study.

We have deeply considered the strategy proposed by the reviewer, comparing “WT cells treated with Activin A/BYL719 and Activin A alone and then compare these identified genes with those shown in Figure 5E” and/or comparing “R206H cells treated with Activin A/BYL719 to WT cells treated with BMP6/BYL719”. While we have discussed why we do not consider appropriate the first comparison proposed, there are a number of reasons why we are not confident that the second comparison would provide a straightforward conclusion.

Regarding the second suggested comparison already in Main point 1, we consider that it would provide confounding results due to all the arguments detailed in Main point 1. Regarding the first suggested comparison, we also consider that it would provide confounding results. There are several reasons why we do not consider that the genes only found in the RH comparison can be confidently considered genes that are only affected by BYL719 in RH cells.

First, the effect of BYL719 in an osteogenic-prone sample (for example, RH-ActA) is higher than the effect that we can observe in absence of this activation (for example, WT-ActA), as observed in the higher number of significantly downregulated genes in RH ActA BYL vs RH ActA comparison, compared to WT ActA BYL vs WT ActA. Similar results are observed in figure 3C, where the expressions of the genes are significantly inhibited in RH ActA compared to RH ActA BYL. This inhibition is not significantly observed in in WT ActA compared to WT ActA BYL because the osteogenic expression of these genes is already very weak in the absence of ACVR1 R206H. This weak signaling of pSMAD1/5 in the absence of osteogenic signaling (RH without ligand or, especially, WT with Activin A) has already been described (Ramachandran et al. MID: 34003511). Therefore, even though the inhibition is present in both comparisons, as observed in figure 6C, the extent of the observed effect is different. Second, we are comparing a different number of DEGs for each comparison between them. If we compare the 67 downregulated genes from one comparison and 38 downregulated genes from the other comparison, the unequal list size may inflate the number of unique genes in the group with more downregulated genes. To prove these concerns, we performed the comparison that the reviewer suggested and we found, for example, that amongst the 38 differentially downregulated ossification genes in (WT_ActA_BYL vs WT_ActA) and 67 differentially downregulated ossification genes in (RH_ActA_BYL vs RH_ActA), 39 genes were only found in the RH comparison, while 10 were only found in the WT comparison, and 28 were found in both.

These effects are present, for example, when studying the ID genes, well-known downstream mediators of BMP signaling. In this case, *ID1* is downregulated in both comparisons, while *ID2*, *ID3*, and *ID4*, are downregulated only in the RH-group, despite the fact that all *ID1*, *ID2*, *ID3*, and *ID4* are similarly regulated and increase their expression with similar time curves upon BMP signaling activation (Yang et al., PMID: 23771884). Therefore, we consider that the comparisons proposed will not help us to identify specific BYL719-regulated genes relevant for FOP and/or ACVR1 R206H signaling. Again, we consider that BYL719 effect is not specific of FOP cells. Our results show that regardless of the ligand (Activin A or BMP6) and regardless of the ACVR1 mutation (WT, relevant for non-genetic heterotopic ossifications or RH, relevant for FOP), BYL719 can inhibit the expression of the genes linked to ossification and osteoblast differentiation, which could be important for the treatment of FOP and non-genetic heterotopic ossifications.

Point 4: The data in Figure 7 are not relevant to the aim of this study because the cell lines used in these experiments did not have ACVR1/R206H mutations. The authors mentioned that BMP6 is a ligand for ACVR1 and, therefore, these experiments reflect the situation of inflammatory cells in FOP. This is inappropriate and not rational. As mentioned above, the effect of Activin A on FOP cells is not identical to the effect of BMP6 in wild-type cells. The data in Figure 7 indicated that the effect of BYL719 is unrelated to the presence of BMP6, clearly demonstrating that these experiments are not related to the activation of ACVR1. In the gene expression analyses, almost all genes showed no changes with the addition of BMP6. Only TGF and CCL2 showed upregulation in THP1 cells, and the treatment with BYL719 failed to inhibit the effect of BMP6, suggesting that these experiments merely demonstrate the effect of BYL719 on inflammatory cells irrespective of the presence of the HO signal.

We consider that Figure 7 is relevant to the aim of this study. As shown in Fig. 8, treatment of FOP mice with BYL719 led to a decreased recruitment of immune cells within the FOP lesions, suggesting a direct effect of BYL719 in immune cells. This is very relevant for the FOP pathology, since flare-ups have been linked with inflammatory episodes since the very early characterization of the disease (Mejias-Rivera et al., PMID: 38672135). Given the technical difficulties to transduce THP1, RAW264 and HMC1 cell lines with lentiviral particles carrying ACVR1 R206H, we decided to partially recapitulate ACVR1 R206H activation with recombinant BMP6 and to test the effect of BYL719 in these conditions. In these models, we found that BYL719 inhibited the expression of key genes driving immune cell activation, in a cell-type and ligand independent manner. To clarify this rationale, we have swapped Figures 7 and 8 and adjusted our conclusions accordingly. We have softened our interpretations, emphasizing the absence of the ACVR1 R206H mutant receptor in these experiments.